# A Faster Decentralized Algorithm for Nonconvex Minimax Problems

**Wenhan Xian, Feihu Huang, Yanfu Zhang, Heng Huang**

Electrical and Computer Engineering, University of Pittsburgh, Pittsburgh, PA 15213

wex37@pitt.edu, huangfeihu2018@gmail.com, yaz91@pitt.edu, heng.huang@pitt.edu

## Abstract

In this paper, we study the nonconvex-strongly-concave minimax optimization problem on decentralized setting. The minimax problems are attracting increasing attentions because of their popular practical applications such as policy evaluation and adversarial training. As training data become larger, distributed training has been broadly adopted in machine learning tasks. Recent research works show that the decentralized distributed data-parallel training techniques are specially promising, because they can achieve the efficient communications and avoid the bottleneck problem on the central node or the latency of low bandwidth network. However, the decentralized minimax problems were seldom studied in literature and the existing methods suffer from very high gradient complexity. To address this challenge, we propose a new faster decentralized algorithm, named as DM-HSGD, for nonconvex minimax problems by using the variance reduced technique of hybrid stochastic gradient descent. We prove that our DM-HSGD algorithm achieves stochastic first-order oracle (SFO) complexity of $O(\kappa^3 \epsilon^{-3})$ for decentralized stochastic nonconvex-strongly-concave problem to search an $\epsilon$-stationary point, which improves the exiting best theoretical results. Moreover, we also prove that our algorithm achieves linear speedup with respect to the number of workers. Our experiments on decentralized settings show the superior performance of our new algorithm.

## 1 Introduction

Minimax optimization has enormous applications in machine learning tasks such as Generative Adversarial Net (GAN) [8], adversarial training [26] and multi-agent reinforcement learning [43]. Specifically, in minimax optimization, variable $x$ aims to minimize a payoff loss function $f(x, y)$ : $\mathbb{R}^{d_1} \times \mathbb{R}^{d_2} \to \mathbb{R}$ while variable $y$ tries to maximize the loss, which can be formulated as

$$\min_{x \in \mathcal{X}} \max_{y \in \mathcal{Y}} f(x, y), \tag{1}$$

where $\mathcal{X} \subseteq \mathbb{R}^{d_1}$ and $\mathcal{Y} \subseteq \mathbb{R}^{d_2}$. In the past a few decades, there are plenty of works to study minimax optimization problem in a variety of research fields and many methods have been developed. The most intuitive solution is Gradient Descent Ascent (GDA) algorithm [6, 29] with equal stepsize $\eta_x = \eta_y$. Asymptotic and nonasymptotic convergence analysis has been provided when $f$ is convex in $x$ and concave in $y$. Recently, many deterministic and stochastic gradient algorithms for nonconvex-strongly-concave and nonconvex-concave problems were proposed. Some algorithms improve the performance of vanilla GDA method by adopting different stepsize on $x$ and $y$, such as [10, 19], where the stepsize of $y$ is typically larger than the stepsize of $x$. Some algorithms update $x$ and $y$ at different frequency, such as [14, 25, 32]. These kind of algorithms usually involve a nested loop structure that updates $y$ more frequently than $x$ to make $f(x, y)$ close to function $\Phi(x)$, which is defined by

$$\Phi(x) = \max_{y \in \mathcal{Y}} f(x, y). \tag{2}$$

35th Conference on Neural Information Processing Systems (NeurIPS 2021).

As more large-scale machine learning problems are arising, distributed training becomes a popular and crucial framework because of its ability and efficiency to deal with large data. It is desired to generalize minimax optimization to distributed training to solve large-scale minimax problems. In distributed optimization, the original centralized optimization suffers from a bottleneck communication problem, *i.e.* the communication traffic on the busiest central node, especially when the network is large [18, 51]. To tackle this communication issue, decentralized optimization was proposed and has emerged as a promising technique. It is a kind of distributed machine learning training paradigm that does not rely on the centralized network topology. Different worker nodes collaboratively utilize their own local data to implement large-scale training tasks and at each iteration they only have to communicate with their neighbors. Decentralized algorithms have been shown to enhance the communication efficiency by avoiding the communication overhead problem. Decentralized methods are also advantageous when the network suffers from communication restriction or has low bandwidth between some nodes and the central node. Besides, it is also an essential method in some situations where data are geographically distributed and centralized data processing is not available or there are concerns to preserve data privacy [48].

Recently many works were proposed to improve the performance of decentralized training. D-PSGD [18] theoretically justifies the potential advantage of decentralized algorithm. $D^2$ [38] improves the convergence rate to outperform D-PSGD by eliminating the influence of data variance among different workers. D-SPIDER-SFO [33] incorporates $D^2$ and SPIDER [7, 44], which is a kind of variance reduction technique [15], to further reduce the gradient complexity. DQSFW [45] studies decentralized constrained problem with Frank-Wolfe method. GT-HSGD [46] extends hybrid stochastic gradient descent to decentralized setting, which is a variance-reduced approach that does not compute mega batch periodically. However, the decentralized minimax optimization is still very limited and existing methods suffer from very high gradient complexity [21, 41]. Thus, we are motivated to design an accelerated decentralized algorithm for minimax problems.

In this paper, thus, we propose a faster Decentralized Minimax Hybrid Stochastic Gradient Descent (DM-HSGD) algorithm to solve the following decentralized stochastic minimax optimization problem:

$$\min_{x \in \mathbb{R}^{d_1}} \max_{y \in \mathcal{Y}} f(x, y) = \frac{1}{n} \sum_{i=1}^{n} f_i(x, y), \quad f_i(x, y) := \mathbb{E}_{\xi^{(i)} \sim D_i} F_i(x, y; \xi^{(i)}) \tag{3}$$

where $n$ is the number of worker nodes, $\mathcal{Y}$ is a convex set. Here the local component objective function $F_i(x, y; \xi^{(i)})$ is $L$-smooth, nonconvex in $x$, and strongly-concave in $y$. $D_i$ is the data distribution on the $i$-th node. In this paper, the data distribution can be non-identical. Random variable $\xi^{(i)}$ is an index sampled from the local data. We summarize our contributions as follows:

(1) In this paper, we propose a new accelerated decentralized stochastic first-order algorithm, named as DM-HSGD, to solve the decentralized nonconvex-strongly-concave minimax optimization problems. Our algorithm is the first stochastic gradient algorithm to solve general decentralized minimax problem on non-identical distributed data with theoretical guarantees. Besides, our algorithm does not require large batch size or nested loop which makes it more practical and efficient to implement.

(2) We provide a completed proof to guarantee the convergence of our algorithm to solve decentralized stochastic minimax optimization. Under nonconvex-strongly-concave condition, our algorithm obtains SFO complexity of $O(\kappa^3 \epsilon^{-3})$ to search an $\epsilon$-stationary point of function $\Phi(x) = \max_{y \in \mathcal{Y}} f(x, y)$. This result is faster than the complexity of previous decentralized minimax algorithms [21, 41]. Moreover, we also prove that our method achieves linear speedup as the number of workers $n$ increases, which verifies its ability to solve large-scale problems.

The rest of this paper will be organized as follows. In Section 2, we will introduce related works. In Section 3, we will introduce our new DM-HSGD algorithm. In Section 4, we will show the main theorems of convergence and complexity analysis. In Section 5, we will discuss our experimental results, and Section 6 will conclude the paper.

Table 1: Comparison of Related Algorithms for Minimax Optimization

| Name | SFO | Decentralized | Stochastic | Implementation | Reference |
|---|---|---|---|---|---|
| SGDA | $O(\kappa^3\epsilon^{-4})$ | $\times$ | $\checkmark$ | single-loop | [19] |
| SGDmax | $O(\kappa^3\epsilon^{-4}\log(\frac{1}{\epsilon}))$ | $\times$ | $\checkmark$ | double-loop | [19] |
| SREDA | $O(\kappa^3\epsilon^{-3})$ | $\times$ | $\checkmark$ | double-loop | [25] |
| Acc-MDA | $O(\kappa^3\epsilon^{-3})$ | $\times$ | $\checkmark$ | single-loop | [11] |
| DPOSG | $O(\epsilon^{-12})$ | $\checkmark$ (iid) | $\checkmark$ | single-loop | [21] |
| GT/DA | $O(N\epsilon^{-2}\log(\frac{1}{\epsilon}))$ | $\checkmark$ (non-iid) | $\times$ | double-loop | [41] |
| DM-HSGD | $O(\kappa^3\epsilon^{-3})$ | $\checkmark$ (non-iid) | $\checkmark$ | single-loop | Ours |

## 2 Related Works

### 2.1 Centralized Minimax Optimization

In recent years, many algorithms for solving minimax optimization were proposed, and the majority of them were studied under the nonconvex-strongly-concave condition. SGDmax [14] is a double loop algorithm that achieves SFO complexity of $O(\kappa^3\epsilon^{-4}log(1/\epsilon))$ where $\kappa = L/\mu$ is the condition number. Proximally Guided Stochastic Mirror Descent and Variance Reduction (PGSMD/PGSVRG) [34] are double loop algorithms that achieve SFO complexity of $O(\kappa^3\epsilon^{-4})$ for stochastic problem and $O(\kappa^2 N\epsilon^{-2})$ for finite-sum problem where $N$ is the number of samples. Multistep GDA (MGDA) [32] is a double loop algorithm and HiBSA [23] is a single loop algorithm. Both MGDA and HiBSA are deterministic hence they can only solve finite-sum problems. Both of them achieve SFO complexity of $O(\kappa^4 N\epsilon^{-2})$. Proximal Dual Implicit Accelerated Gradient (ProxDIAG) is a deterministic triple loop algorithm whose SFO complexity for finite-sum problem is $O(\kappa^{1/2} N\epsilon^{-2})$.

SGDA [19], Stochastic Recursive gradiEnt Descent Ascent (SREDA) [25], and Hybrid Variance-Reduced SGD [40] are more related to our work. SGDA is a single loop algorithm to solve nonconvex-strongly-concave and nonconvex-concave minimax problems. For nonconvex-strongly-concave problem, it requires $O(\kappa^3\epsilon^{-4})$ SFO complexity to find an $\epsilon$-stationary point of $\Phi(x)$. In this paper, we will prove that our method achieves a better SFO complexity.

SREDA [25] is a double loop algorithm that achieves $O(\kappa^3\epsilon^{-3})$ SFO complexity. It accelerates SGDA by using SPIDER, which is a variance reduction technique and utilizes the newest gradient information [7, 30]. SREDA also involves a separated initialization algorithm called PiSARAH [31] to ensure the convergence. More recently, [13] proposed an efficient mirror descent ascent algorithm for nonconvex-strongly-concave minimax optimization with nonsmooth regularization based on Bregman distance and variance reduced technique of SPIDER. In our paper, we use another variance-reduced technique named STORM or hybrid stochastic gradient descent [3] to accelerate the algorithm. We will discuss the challenges of using SPIDER on decentralized settings in Section 3. Different from SREDA, our method only requires a large batch at the first iteration. Except the first iteration, we can use either a single sample or a mini-batch to calculate the stochastic gradient. However, SREDA loads a mega-batch with size $O(\epsilon^{-2})$ periodically (every $q$ iterations) and needs $O(\epsilon^{-1})$ gradient oracles at each iteration, which is not practical for large-scale problems. Besides, the maximizer in SREDA is a nested loop to update variable $y$ and if we count the loop of SPIDER then SREDA is actually a triple algorithm. On the contrary, there is no nested loop in our DM-HSGD, which makes our method more efficient and convenient to implement. Moreover, unlike SREDA, our method does not require a separated initialization algorithm to calculate a precise initial value for $y$.

Hybrid Variance-Reduced SGD algorithm also takes advantage of hybrid stochastic gradient descent to accelerate minimax optimization. For example, [40, 11] applied the Hybrid Variance-Reduced SGD to minimax problems. More recently, [9, 12] proposed some efficient adaptive gradient descent ascent methods for nonconvex-strongly-concave minimax optimization based on momentum techniques including Hybrid Variance-Reduced SGD.

### 2.2 Decentralized Minimax Optimization

At decentralized setting, most minimax algorithms were proposed for convex-concave problem [17, 28]. In [22] a nonconvex-nonconcave algorithm DPPSP was proposed. However, it is not

gradient-based and the closed-form solution to the subproblem is not ensured in our problem. Hence we will not discuss it in this paper. Decentralized Parallel Optimistic Stochastic Gradient (DPOSG) [21] is the first algorithm applicable to a general decentralized minimax problem with theoretical guarantees. It is a single loop minimax algorithm that generalizes Optimistic Stochastic Gradient (OSG) [2] to decentralized training. However, DPOSG has some obvious drawbacks. The first one is that the gradient complexity $O(\epsilon^{-12})$ is too high and we are motivated to design a faster algorithm. The second one is that DPOSG only works in the case where the data distribution is identical. When the data distribution is non-identical, the Lemma 3 in [21] is not satisfied. Actually the assumption of identical data distribution is not satisfied at most decentralized training tasks. Thus, in this paper, we do not use this assumption.

More recently, [41] studied decentralized nonconvex-strongly-concave minimax problems and proposed a double loop deterministic Gradient Tracking/Descent-Ascent algorithm which extends the vanilla GDA to decentralized setting and combines it with gradient tracking. It achieves a gradient complexity of $O(\epsilon^{-2})$. However, in large-scale machine learning tasks such as deep neural network, generally the full gradient is unavailable and the application of deterministic algorithms is very restricted. If we convert Gradient Tracking/Descent-Ascent to stochastic gradient version, the SFO complexity should be at least $O(\epsilon^{-4})$, which is the same result as SGD in nonconvex optimization. Under the same conditions, our new algorithm achieves a better SFO complexity of $O(\epsilon^{-3})$.

[24] studied decentralized reinforcement learning problem based on distributed constrained Markov decision process model and proposed a decentralized policy gradient optimization method named Safe Dec-PG, which achieves SFO complexity of $O(\epsilon^{-4})$. However, the problem studied in [24] has a special form that is linear in $y$. In this paper, we focus on general minimax problem. [1] is a simultaneous work of our work that studies a more general decentralized variational inequality problem with higher complexity. We summarize the comparison of related algorithms for general minimax optimization in Table 1. For decentralized algorithms DPOSG, GT/DA, and DM-HSGD, we also discuss whether they can converge on non-identical distributed data.

## 3 Proposed New Algorithm

### 3.1 Preliminaries

Before we propose our algorithms, we will introduce the notations used in this paper and some important concepts. We use lower case $x_t^{(i)}$ and $y_t^{(i)}$ to represent the column vector parameters on $i$-th worker node. We use upper case $X_t$ and $Y_t$ to represent the $n$-column matrix formed by $x_t^{(i)}$ and $y_t^{(i)}$ respectively, which means $X_t = [x_t^{(1)}, x_t^{(2)}, \ldots, x_t^{(n)}]$ and $Y_t = [y_t^{(1)}, y_t^{(2)}, \ldots, y_t^{(n)}]$. Column vectors $u_t^{(i)}$, $v_t^{(i)}$, $g_t^{(i)}$ and $h_t^{(i)}$ are gradient estimators used in our algorithms. Upper case $U_t$, $V_t$, $G_t$ and $H_t$ are matrices of which the $i$-th column is $u_t^{(i)}$, $v_t^{(i)}$, $g_t^{(i)}$ and $h_t^{(i)}$ respectively. Lower case with a bar represents the mean vector. Upper case with a bar represents the matrix that each column is the mean vector. For example, $\bar{x}_t = \frac{1}{n} \sum_{i=1}^{n} x_t^{(i)}$ and $\bar{X}_t = [\bar{x}_t, \bar{x}_t, \ldots, \bar{x}_t]$. We define the optimal maximum value of $y$ as:

$$y^*(\cdot) = \arg\max_{y \in \mathcal{Y}} f(\cdot, y), \quad \hat{y}_t = \arg\max_{y \in \mathcal{Y}} f(\bar{x}_t, y) \tag{4}$$

Note that when $f$ is strongly-concave in $y$, $\hat{y}_t$ is unique. We also define:

$$\delta_t = \|\hat{y}_t - \bar{y}_t\|^2 \tag{5}$$

Bold number $\mathbf{0}$ and $\mathbf{1}$ are $n \times 1$ column vectors that each entry is 0 and 1, respectively. For matrices, we use $\|\cdot\|_F$ to denote Frobenius norm and $\|\cdot\|_2$ to denote spectral norm. We use $\nabla_x$ and $\nabla_y$ to denote the partial derivative with respect to $x$ and $y$.

Mixing matrix $W$ represents the weights of averaging among the communication network topology. It is doubly stochastic which satisfies:

$$W\mathbf{1} = W^T\mathbf{1} = \mathbf{1} \tag{6}$$

We should notice that here matrix $W$ is not assumed to be symmetric so that the communication network is not restricted to undirected graph.

---

**Algorithm 1** DM-HSGD

---

**Input**: mixing matrix $W$, initial value $x_0^{(i)} = x_0$, $y_0^{(i)} = y_0$, $v_{-1}^{(i)} = g_{-1}^{(i)} = \mathbf{0}$, $u_{-1}^{(i)} = h_{-1}^{(i)} = \mathbf{0}$
**Parameter**: stepsize $\eta_x$, $\eta_y$, weight $\beta_x$, $\beta_y$, batch size $b_0$, iteration $T$
**Output**: $\bar{x}_\zeta$, where $\zeta$ is chosen randomly from $\{1, 2, \cdots, T\}$

1: On i-th node:
2: **for** $t = 0, 1, \ldots, T - 1$ **do**
3:    **if** $t = 0$ **then**
4:       $g_t^{(i)} = \nabla_x F_i(x_t^{(i)}, y_t^{(i)}; \xi_{x,t}^{(i)}), \quad |\xi_{x,t}^{(i)}| = b_0$
5:       $h_t^{(i)} = \nabla_y F_i(x_t^{(i)}, y_t^{(i)}; \xi_{y,t}^{(i)}), \quad |\xi_{y,t}^{(i)}| = b_0$
6:    **else**
7:       $g_t^{(i)} = \nabla_x F_i(x_t^{(i)}, y_t^{(i)}; \xi_t^{(i)}) + (1 - \beta_x)(g_{t-1}^{(i)} - \nabla_x F_i(x_{t-1}^{(i)}, y_{t-1}^{(i)}; \xi_t^{(i)}))$
8:       $h_t^{(i)} = \nabla_y F_i(x_t^{(i)}, y_t^{(i)}; \xi_t^{(i)}) + (1 - \beta_y)(h_{t-1}^{(i)} - \nabla_y F_i(x_{t-1}^{(i)}, y_{t-1}^{(i)}; \xi_t^{(i)}))$
9:    **end if**
10:   Communicate with neighbors and update gradient estimator as follows
11:   $v_t^{(i)} = \sum_{j=1}^{n} w_{ij}(v_{t-1}^{(j)} + g_t^{(j)} - g_{t-1}^{(j)})$
12:   $u_t^{(i)} = \sum_{j=1}^{n} w_{ij}(u_{t-1}^{(j)} + h_t^{(j)} - h_{t-1}^{(j)})$
13:   Communicate with neighbors and update model parameter as follows
14:   $x_{t+1}^{(i)} = \sum_{j=1}^{n} w_{ij}(x_t^{(j)} - \eta_x v_t^{(j)})$
15:   $y_{t+\frac{1}{2}}^{(i)} = \sum_{j=1}^{n} w_{ij}(y_t^{(j)} + \eta_y u_t^{(j)}), \quad y_{t+1}^{(i)} = P_{\mathcal{Y}}(y_{t+\frac{1}{2}}^{(i)})$
16: **end for**

---

### 3.2 Decentralized Minimax Hybrid Stochastic Gradient Descent

In this subsection, we introduce our new Decentralized Minimax Hybrid Stochastic Gradient Descent (DM-HSGD) algorithm. Our algorithm is a single loop minimax algorithm (summarized in Algorithm 1) which does not contain a nested loop structure.

The initial points of different nodes are the same, *i.e.* $x_0^{(i)} = x_0$ and $y_0^{(i)} = y_0$. $g_t^{(i)}$ and $h_t^{(i)}$ are the gradient estimators with respect to $x$ and $y$ on $i$-th node. $g_t^{(i)}$ and $h_t^{(i)}$ are computed in the same way as STORM [3]. When $t = 0$, we load a large batch with size $b_0$ to calculate the stochastic gradient (lines 4 and 5 in Algorithm 1). When $t > 0$, we can use either a single sample or a mini-batch to calculate the gradient (lines 7 and 8 in Algorithm 1). $g_t^{(i)}$ can also be written as

$$g_t^{(i)} = \beta_x \nabla_x F_i(x_t^{(i)}, y_t^{(i)}; \xi_t^{(i)}) + (1 - \beta_x)\Big(g_{t-1}^{(i)} - \nabla_x F_i(x_{t-1}^{(i)}, y_{t-1}^{(i)}; \xi_t^{(i)}) + \nabla_x F_i(x_t^{(i)}, y_t^{(i)}; \xi_t^{(i)})\Big) \quad (7)$$

which is a linear combination of the gradient estimators of stochastic gradient descent (the first part) and SPIDER (the second part). As we have mentioned, SPIDER is a variance-reduced method that utilizes the newest gradient information. Thus, estimator Eq. (7) is also called hybrid stochastic gradient descent. It is the same with $h_t^{(i)}$. Then each worker communicates with their neighbors to compute gradient estimator $v_t^{(i)}$ and $u_t^{(i)}$. Here we use gradient tracking [5, 47] to reduce the consensus error (lines 11 and 12 in Algorithm 1). We will discuss why gradient tracking is necessary in our method at next subsection. After we obtain $u_t^{(i)}$ and $v_t^{(i)}$, each worker communicates with their neighbors again and updates the model parameters $x$ and $y$. Here $P_{\mathcal{Y}}(\cdot)$ represents the projection onto convex set $\mathcal{Y}$. In the theoretical analysis, we define $Y_{-\frac{1}{2}} = Y_0$.

### 3.3 Discussions on STORM and Gradient Tracking

In this subsection, we will discuss the intuition of our algorithm and explain why we choose STORM and gradient tracking rather than generalizing SREDA for decentralized setting. The first reason is that SREDA requires large batch or full batch periodically, which is expensive and even unavailable. Besides, there are too many nested loops in SREDA and it is not efficient or convenient. From the view of theoretical analysis, normalization or projection are likely to cause divergence in decentralized training on non-identical data distribution, which is indicated by the following Example 1. Therefore, in the circumstance of this paper, SPIDER will probably not converge to a stationary point. Besides,

SREDA adopts smaller stepsize at the beginning and larger stepsize at the end when $\|v_t\|$ becomes small enough. However, when the data distribution is non-identical, $\|v_t\|$ may not tend to 0 and the stepsize of SREDA will probably always keep small. In contrast, STORM can avoid these issues and we use STORM to accelerate the decentralized minimax algorithm.

In the standard decentralized framework D-PSGD [18], the consensus error satisfies $\|X_t - \bar{X}_t\|_F \leq O(\epsilon)$ when the stepsize $\eta$ is $O(\epsilon)$ and $t$ is large enough. The following Example 2 is a simple example to show that this bound is tight and there are cases where consensus error $\|X_t - \bar{X}_t\|_F$ is exactly $\Theta(\eta)$ when the data distribution is non-identical. However, according to the analysis of STORM [3] without gradient tracking, the error term $e_t = \bar{g}_t - \nabla_x f(\bar{x}_t, \bar{y}_t)$ between the averaged update direction and the correct direction is supposed to satisfy:

$$\|e_t\|^2 \leq (1 - \beta_x)\|e_{t-1}\|^2 + O(\eta_x^4). \tag{8}$$

Nevertheless, the consensus error $\|X_t - \bar{X}_t\|_F^2$ is only $O(\eta_x^2)$ and cannot be as small as $O(\eta_x^4)$ if there is no gradient tracking. Therefore, to inherit the analysis framework of STORM, the gradient tracking in our algorithm is essential.

**Example 1.** *Assume $f(x) = f_1(x) + f_2(x)$, where $x = (a, b) \in R^2$. $f_1(x) = a$ and $f_2(x) = \sqrt{3}b$ are defined on two different nodes. Let $W$ be the uniform weighted mixing matrix. We can compute $v_1 = (1, 0)$ and $v_2 = (0, \sqrt{3})$. The ideal averaged gradient direction is $v^* = (1/2, \sqrt{3}/2)$. However, if we do normalization before making consensus, the obtained gradient estimator is $v = (1/2, 1/2)$, which is deviated from $v^*$.*

**Example 2.** *Suppose there are two sequences $\{p_t\}$ and $\{q_t\}$ defined on two different nodes with $p_0 = q_0$. They are updated by $p_{t+\frac{1}{2}} = p_t - \eta a$ and $q_{t+\frac{1}{2}} = q_t - \eta b$ at each iteration respectively where $a$ and $b$ are fixed gradient directions. As data distribution is non-identical, we have $a \neq b$. Assume the mixing matrix is*

$$W = \begin{bmatrix} 2/3 & 1/3 \\ 1/3 & 2/3 \end{bmatrix}$$

*Then we have*

$$p_{t+1} - q_{t+1} = \frac{1}{3}(p_t - q_t) - \frac{\eta}{3}(a - b) = \frac{1}{3^{t+1}}(p_0 - q_0) - \eta(\sum_{s=1}^{t+1} \frac{1}{3^s})(a - b) = \frac{\eta}{2}(1 - \frac{1}{3^{t+1}})(b - a) \tag{9}$$

*Therefore, $\lim_{t\to\infty}\|p_t - q_t\| = \frac{\eta}{2}\|a - b\|$.*

## 4  Convergence Analysis

In this section, we will show the main theorems of our convergence analysis. The theoretical results show that the SFO complexity of our algorithm is $O(\kappa^3 \epsilon^{-3})$, which is the same as the best result in centralized minimax problem [25]. First we will introduce the following assumptions.

**Assumption 1.** *(Lipschitz Gradient). Each component function $F_i(x, y; \xi)$ is $L$-smooth, which means there exists a constant $L$ such that for any $(x, y)$ and $(x', y')$, we have*

$$\|\nabla F_i(x, y; \xi) - \nabla F_i(x', y'; \xi)\|^2 \leq L^2(\|x - x'\|^2 + \|y - y'\|^2)$$

**Assumption 2.** *(Bounded Variance). The gradient of each component function $F_i(x, y; \xi)$ is an unbiased estimator of $\nabla f_i(x, y)$ and has bounded variance, i.e.,*

$$\mathbb{E}\|\nabla F_i(x, y; \xi) - \nabla f_i(x, y)\|^2 \leq \sigma < +\infty$$

**Assumption 3.** *(Lower Bound). The function $\Phi(\cdot)$ is lower bounded, i.e., $\inf_x \Phi(x) = \Phi^* > -\infty$.*

**Assumption 4.** *(Spectral Gap). The doubly stochastic matrix $W$ satisfies $\|W - \frac{\mathbf{1}\mathbf{1}^T}{n}\|_2 = \lambda \in [0, 1)$.*

**Assumption 5.** *(Strongly Concave). The function $f_i(x, y)$ is $\mu$-strongly-concave in $y$. That is, there exists a constant $\mu > 0$, for any $x, y$ and $y'$, we have*

$$f_i(x, y) \leq f(x, y') + \langle \nabla_y f(x, y'), y - y' \rangle - \frac{\mu}{2}\|y - y'\|^2$$

These are very common and mild assumptions that are frequently assumed in previous works. Assumptions 1, 2 and 3 are also used in minimax methods [25] and [19]. Assumption 4 is used in [46]. Typically, the spectral gap assumption is stated as $W$ is symmetric and $|\lambda_2| < 1$, $|\lambda_n| < 1$ where $\lambda_1 \geq \lambda_2 \geq \cdots \geq \lambda_n$ are the eigenvalues of $W$ [16, 18, 51]. Our Assumption 4 is automatically satisfied if the typical spectral gap assumption holds (see Lemma 16 in [16]). Assumption 5 is the definition of strong concavity.

In nonconvex-strongly-concave problem, we use $\epsilon$-stationary point of $\Phi(x)$, *i.e.* $\|\nabla\Phi(x)\| \leq \epsilon$ as the convergence criterion. From Lemma 4.3 in [19], we know $\Phi(x)$ is differentiable and $(L+\kappa L)$-smooth and $y^*(\cdot)$ is $\kappa$-Lipschitz, which means $\|y^*(x_1) - y^*(x_2)\| \leq \kappa\|x_1 - x_2\|$ for any $x_1, x_2 \in \mathbb{R}^{d_1}$. Furthermore, we have:

$$\nabla\Phi(\bar{x}_t) = \nabla_x f(\bar{x}_t, \hat{y}_t) + \nabla_y f(\bar{x}_t, \hat{y}_t) \cdot \partial y^*(\bar{x}_t) = \nabla_x f(\bar{x}_t, \hat{y}_t) \tag{10}$$

since $\nabla_y f(\bar{x}_t, \hat{y}_t) = 0$. This criterion is broadly used in the analysis of nonconvex-strongly-concave minimax optimization [19, 39]. Now we will provide the main theorems of our convergence analysis. Completed proof can be found in the Supplementary Material.

**Theorem 1.** *Let Assumptions 1 to 5 hold. When parameters* $\beta_x = \frac{\epsilon \min\{1, n\epsilon\}}{20}$, $\beta_y = \frac{\epsilon \min\{1, n\epsilon\}}{500\kappa^2}$, $\eta_x = \frac{(1-\lambda)^2 \min\{1, n\epsilon\}}{2000\kappa^3 L}$, $\eta_y = \frac{(1-\lambda)^2 \min\{1, n\epsilon\}}{500\kappa L}$, $b_0 = \frac{400}{\min\{1, n\epsilon\}}$, $T = \frac{4000\kappa^3 \epsilon^{-2}}{(1-\lambda)^2 \min\{1, n\epsilon\}}$, *our Algorithm 1 satisfies*

$$\frac{1}{T}\sum_{t=0}^{T-1}\mathbb{E}\|\nabla\Phi(\bar{x}_t)\|^2 \leq L(\Phi(x_0) - \Phi^*)\epsilon^2 + \sigma^2\epsilon^2 + L^2\delta_0\epsilon^2 + \frac{\epsilon^2}{n}\sum_{i=1}^{n}\mathbb{E}\|\nabla_x f_i(x_0, y_0)\|^2$$

$$+ \frac{\epsilon^2}{n}\sum_{i=1}^{n}\mathbb{E}\|\nabla_y f_i(x_0, y_0)\|^2 \tag{11}$$

**Corollary 1.** *When the parameters are defined as Theorem 1, we can see* $\frac{1}{T}\sum_{t=0}^{T-1}\mathbb{E}\|\nabla\Phi(\bar{x}_t)\|^2 \leq O(\epsilon^2)$. *Therefore, if* $n \leq O(\epsilon^{-1})$, *the SFO complexity of Algorithm 1 is* $O(\kappa^3\epsilon^{-3})$. *If* $n > O(\epsilon^{-1})$, *the SFO complexity is* $O(\kappa^3 n\epsilon^{-2})$. *Besides, from the proof of Theorem 1 we can see error* $\|\bar{y}_t - y^*(\bar{x}_t)\|^2$ *is also bounded by the right side of Eq. (11).*

Theorem 1 is the theoretical result when $T$ is determined by $\epsilon$. If the number of iteration $T$ is not fixed, we have the following conclusion.

**Theorem 2.** *Let Assumptions 1 to 5 hold. We set the parameters as* $T = \frac{4000\kappa^3 T_0}{(1-\lambda)^2}$, $\beta_x = \frac{n^{1/3}}{20T_0^{2/3}}$, $\beta_y = \frac{n^{1/3}}{500\kappa^2 T_0^{2/3}}$, $\eta_x = \frac{(1-\lambda)^2 n^{2/3}}{2000\kappa^3 T_0^{1/3} L}$, $\eta_y = \frac{(1-\lambda)^2 n^{2/3}}{500\kappa T_0^{1/3} L}$, $b_0 = \frac{T_0^{1/3}}{n^{2/3}}$, *where we suppose* $T_0 \geq 10n^2$. *Then our algorithm satisfies*

$$\frac{1}{T}\sum_{t=0}^{T-1}\mathbb{E}\|\nabla\Phi(\bar{x}_t)\|^2 \leq \frac{L(\Phi(x_0) - \Phi^*) + \sigma^2 + L^2\delta_0}{(nT_0)^{2/3}} + \frac{\frac{1}{n}\sum_{i=1}^{n}\mathbb{E}\|\nabla_x f_i(x_0, y_0)\|^2}{T_0}$$

$$+ \frac{\frac{1}{n}\sum_{i=1}^{n}\mathbb{E}\|\nabla_y f_i(x_0, y_0)\|^2}{T_0} \tag{12}$$

**Corollary 2.** *From Theorem 2, we know* $\frac{1}{T}\sum_{t=0}^{T-1}\mathbb{E}\|\nabla\Phi(\bar{x}_t)\|^2 \leq O(\frac{1}{(nT_0)^{2/3}}) + O(\frac{1}{T_0})$ *when parameters are defined as above. As we suppose* $T_0 \geq O(n^2)$, *the dominating term in the convergence rate is* $O(\frac{1}{(nT_0)^{2/3}})$, *which indicates the linear speedup of our algorithm.*

## 5 Experiments

### 5.1 Robust Logistic Regression

We conduct the experiment of decentralized robust logistic regression[1] task as the first experiment, which was proposed in [49] and was also conducted in the related work [25]. Given dataset

---

[1] https://github.com/TrashzzZ/DM-HSGD

$\{(a_i, b_i)\}_{i=1}^n$, where $a_i \in \mathbb{R}^d$ is the feature and $b_i \in \{-1, 1\}$ is the label, the robust logistic regression problem is formulated as follows:

$$\min_{x \in \mathbb{R}^d} \max_{y \in \Delta_n} f(x, y) = \sum_{i=1}^n y_i l_i(x) - V(y) + g(x) \tag{13}$$

where $y_i$ is the $i$-th component of variable $y$. $l_i(x)$ is the logistic loss function which is defined by $l_i(x) = \log(1 + \exp(-b_i a_i^T x))$. $V(y)$ is a divergence measure defined by $V(y) = \frac{1}{2}\lambda_1 \|ny - \mathbf{1}\|^2$. $\Delta_n$ represents the simplex in $\mathbb{R}^n$, which means

$$\Delta_n = \{y \in \mathbb{R}^n | 0 \le y_i \le 1, \sum_{i=1}^n y_i = 1\} \tag{14}$$

$g(x)$ is a nonconvex regularization with form $g(x) = \lambda_2 \sum_{i=1}^d \frac{\alpha x_i^2}{1 + \alpha x_i^2}$. Following the experimental settings in [25, 49], we let $\lambda_1 = \frac{1}{n^2}$, $\lambda_2 = 0.001$ and $\alpha = 10$ in our experiment.

Table 2: Descriptions of datasets used in our experiment

| Name | a9a | covtype | ijcnn1 | phishing | rcv1 | w8a |
|---|---|---|---|---|---|---|
| N | 32561 | 581012 | 49990 | 11055 | 20242 | 49749 |
| d | 123 | 54 | 22 | 68 | 47236 | 300 |

We conduct our experiment on six real-world training datasets "a9a", "covtype", "ijcnn1", "phishing","rcv1" and "w8a", which can be downloaded from LIBSVM[2] repository. The description of datasets is listed in Table 2 where $N$ is the number of samples and $d$ is the number of features. We implement our code on an MPI cluster where each node is equipped with 12-core Intel Xeon E5-2620 v3 2.40 GHz processor.

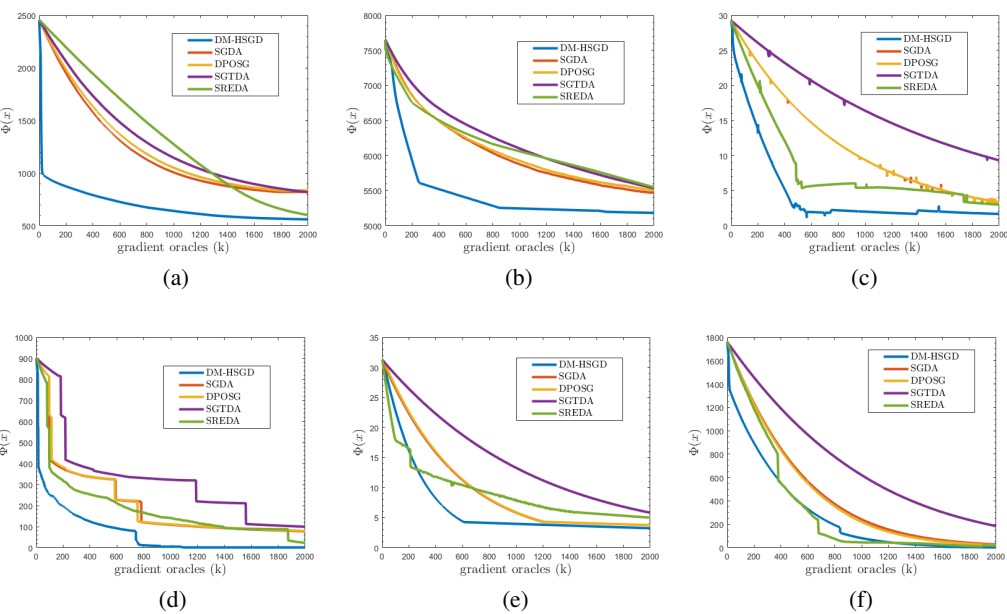

Figure 1: Results of our decentralized robust logistic regression task. Figure (a) to (f) show the value of $\Phi(x)$ with respect to the number of gradient oracles divided by $10^3$. Figure (a), (b), (c), (d), (e) and (f) are experimental results on "a9a", "covtype", "ijcnn1", "phishing", "rcv1" and "w8a" respectively.

We compare our DM-HSGD algorithm with baseline algorithms: SGDA [19], SREDA [25], DPOSG [21], and stochastic Gradient Tracking/Descent Ascent (SGTDA) [41]. We consider the algorithms

---
[2]https://www.csie.ntu.edu.tw/~cjlin/libsvmtools/datasets

for solving stochastic problem. We set the number of worker nodes to $n = 20$ and use the ring-based topology as the communication network. For each algorithm, we grid search the learning rates $\eta_x$ and $\eta_y$ from $\{0.1, 0.01, 0.001, 0.0001\}$. The mini-batch size is set to 20. The number of iterations in the nested loop for double-loop algorithms is set to $K = 5$. For DM-HSGD, we set the batch size of the first iteration to $b_0 = 10000$. $\beta_x$ and $\beta_y$ are set to 0.01. For SREDA, we set $\epsilon = 0.1$ in the factor $\frac{\epsilon}{\|v_t\|}$, period $q = 50$ and large batch size $S_1 = 1000$. We compare the value of $\Phi(x)$ with respect to the number of gradient oracles among different algorithms, which can also be calculated by the projection onto simplex $\Delta_n$. The experimental results are shown in Figure 1. From the experimental results in Figure 1, we can see our new DM-HSGD algorithm converges faster than other baseline algorithms, which verifies the performance of our method.

## 5.2 Policy Evaluation

Our second experiment is the decentralized policy evaluation (PE) task. PE is an important task in reinforcement learning, which aims to estimate the value function of a given policy. The most intuitive and frequently used method for PE is temporal-difference (TD) method that relies on the Bellman equation [4]. However, traditional TD method, which is probably not true gradient descent method as pointed out in [20] and [37], are shown to be unstable in the case of off-policy sampling or nonlinear function approximation. [36] first proposed a method to optimize the objective function of mean-squared projected Bellman error (MSPBE) and MSPBE is proven to achieve asymptotic convergence with arbitrary nonlinear smooth function approximation in [27]. In [42], the MSPBE objective function with nonlinear approximation is converted into a nonconvex-strongly-concave minimax problem by Fenchel's duality. The problem can be formulated as:

$$\min_{\theta} \max_{w} L(\theta, w) = \frac{1}{nN_i} \sum_{i=1}^{n} \sum_{j=1}^{N_i} L_j^{(i)}(\theta, w),$$

$$L_j^{(i)}(\theta, w) = \langle w, [R_i(s_j, a_j) + \gamma V_\theta(s_{j+1}) - V_\theta(s_j)] g_\theta(s_j) \rangle - \frac{1}{2}(w^T g_\theta(s_j))^2 \qquad (15)$$

where $s_j$ is a state and $a_j$ is an action. $R_i$ represents the reward and $\gamma \in (0, 1)$ is the discount factor. $V$ is a value function that maps the state space to a real number. $\theta$ is the parameter to estimate the value function. Function $g_\theta$ is the gradient of $V_\theta$ and parameter $w$ is yield by Fenchel's duality.

Mountaincar [35] is a preliminary task in reinforcement learning. [42] and [43] ran offline PE task of this problem with primal-dual MSPBE, where the objective function is formulated as Eq. (15). Following the experimental settings in [42], we use Sarsa [35] to generate trajectories of transitions $(s_i, a_i, s_{i+1}, r_i)$ with $d$ features and $N = 5000$ samples on each worker node. We parameterize value function $V_\theta$ as a 2-layer neural network with $H$ hidden neurons. We use Sigmoid function as activation and set discount factor to $\gamma = 0.95$. This experiment is run on an MPI cluster where each node is equipped with 12-core Intel Xeon E5-2620 v3 2.40 GHz processor.

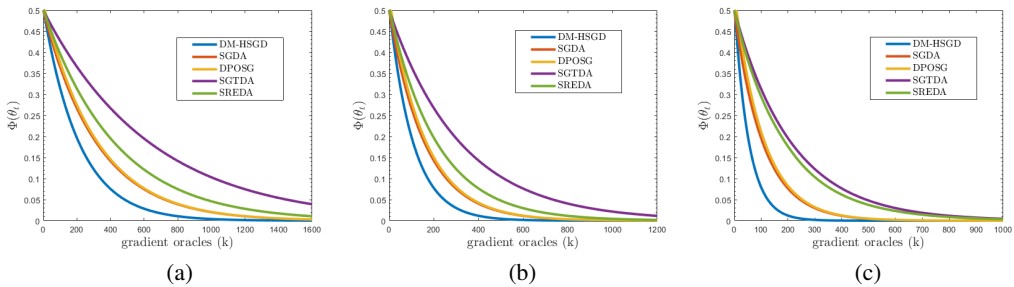

Figure 2: Results of our policy evaluation task. Figures (a), (b) and (c) show the value of $\Phi(\theta)$ with respect to the number of gradient oracles divided by $10^3$. In Figures, (a) $d = 200$, $H = 50$; (b) $d = 300$, $H = 100$; (c) $d = 400$, $H = 200$.

We compare our DM-HSGD algorithm with baseline algorithms: SGDA [19], SREDA [25], DPOSG [21], and stochastic Gradient Tracking/Descent Ascent (SGTDA) [41]. We also consider algorithms

for solving stochastic problem. We set the number of worker nodes to $n = 20$. We also use a ring-based topology with uniform weights as the communication network in this task. For each algorithm, we grid search the learning rates $\eta_x$ and $\eta_y$ from $\{0.1, 0.01, 0.001, 0.0001\}$. The mini-batch size is set to 20. The number of iterations in the nested loop for double-loop algorithms is set to $K = 5$. For DM-HSGD, we set the batch size of the first iteration to $b_0 = 2500$. $\beta_x$ and $\beta_y$ are set to 0.01. For SREDA, we set $\epsilon = 0.1$ in the factor $\frac{\epsilon}{\|v_t\|}$, period $q = 50$ and large batch size $S_1 = 1000$. We compare the value of $\Phi(\theta)$ with respect to the number of gradient oracles among different algorithms, which can be calculated by quadratic optimization. The experimental results are shown in Figure 2.

Figure 2 (a), (b) and (c) show that our DM-HSGD algorithm achieves the fastest convergence regarding the number of gradient oracles. From the experimental result, we can also see that nested loop algorithm for minimax optimization usually consumes more gradient complexity during the training process than single-loop algorithm.

## 6 Conclusion

In this paper, we proposed a novel accelerated decentralized minimax algorithm, Decentralized Minimax Hybrid Stochastic Gradient Descent (DM-HSGD), to solve the stochastic nonconvex-strongly-concave minimax optimization problems. We prove that our new method obtains SFO complexity of $O(\kappa^3 \epsilon^{-3})$ which outperforms the existing results in decentralized minimax optimization and matches state-of-the-art in centralized minimax optimization. Our method also achieves linear speedup with respect to the number workers, which shows its ability to solve large-scale problems. We also conduct experiments on two machine learning tasks, decentralized robust logistic regression and policy evaluation to validate the superior performance of our algorithm. In our future work, we will explore the decentralized nonconvex-concave minimax optimization without the strong concavity so that it can solve a broader range of problems including the loss functions that are linear in $y$. We will probably consider the methods that add a perturbation such as Catalyst [50].

## Acknowledgments and Disclosure of Funding

This work was partially supported by NSF IIS 1845666, 1852606, 1838627, 1837956, 1956002, OIA 2040588.

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
