# A Proof of Convergence Analysis

## A.1 Basic Lemmas

First, we introduce following basic lemmas, which are broadly used in the convergence analysis of optimization algorithms.

**Lemma 1.** *Let vector $X$ be a stochastic variable. Then we have*

$$0 \leq \mathbb{E}\|X - \mathbb{E}X\|^2 = \mathbb{E}\|X\|^2 - \|\mathbb{E}X\|^2 \leq \mathbb{E}\|X\|^2 \tag{16}$$

**Lemma 2.** *Let $X_1, X_2, \cdots, X_n$ be $n$ independent stochastic variables of which the means are $0$. Then we have*

$$\mathbb{E}\|\sum_{i=1}^{n} X_i\|^2 = \sum_{i=1}^{n} \mathbb{E}\|X_i\|^2 \tag{17}$$

**Lemma 3.** *Suppose $A$ and $B$ are two matrices. Then it satisfies*

$$\|AB\|_F \leq \|A\|_2 \|B\|_F \tag{18}$$

## A.2 Important Conclusions

Next, we will propose and prove some conclusions that are important to the proof our main theorems.

**Lemma 4.** *(Lemma 4.3 in paper [19]) $\Phi(x)$ is $(L + \kappa L)$-smooth and $y^*(\cdot)$ is $\kappa$-Lipschitz, which means $\|y^*(x_1) - y^*(x_2)\| \leq \kappa \|x_1 - x_2\|$ for any $x_1, x_2 \in \mathbb{R}^{d_1}$.*

*Proof.* As $y^*(x_1)$ and $y^*(x_2)$ achieve the maximum, we have $\nabla_y f(x_1, y^*(x_1)) = \mathbf{0}$ and $\nabla_y f(x_2, y^*(x_2)) = \mathbf{0}$. Then we have

$$\begin{aligned}
&\|y^*(x_1) - y^*(x_2)\| \\
&\leq \frac{1}{\mu} \|\nabla_y f(x_1, y^*(x_1)) - \nabla_y f(x_1, y^*(x_2))\| \\
&= \frac{1}{\mu} \|\nabla_y f(x_2, y^*(x_2)) - \nabla_y f(x_1, y^*(x_2))\| \leq \frac{L}{\mu} \|x_1 - x_2\| = \kappa \|x_1 - x_2\|
\end{aligned} \tag{19}$$

where the first inequality is derived from $\mu$-strong concavity and the second inequality is derived from $L$-smoothness. Since $\nabla \Phi(x) = \nabla_x f(x, y^*(x))$, from Assumption 1 we get

$$\|\nabla \Phi(x_1) - \nabla \Phi(x_2)\| \leq L \|x_1 - x_2\| + L \|y^*(x_1) - y^*(x_2)\| \leq (L + \kappa L) \|x_1 - x_2\| \tag{20}$$

which implies $\Phi(x)$ is $(L + \kappa L)$-smooth. $\qquad\square$

**Lemma 5.** *When $\eta_y \leq \frac{1}{5L}$ we have following estimation for $\delta_t$.*

$$\begin{aligned}
\sum_{t=0}^{T-1} \delta_t \leq{} & \frac{4\kappa}{L\eta_y} \delta_0 + \frac{18\eta_y}{\mu} \sum_{t=1}^{T-1} (1 - \frac{\mu\eta_y}{4})^{T-t-1} \sum_{s=0}^{t-1} \|\bar{u}_s - \frac{1}{n} \sum_{i=1}^{n} \nabla f_i(x_s^{(i)}, y_s^{(i)})\|^2 + \frac{72\kappa^2}{n} \sum_{t=0}^{T-1} (\|X_t - \bar{X}_t\|_F^2 \\
& + \|Y_t - \bar{Y}_t\|_F^2) + \frac{20\kappa^4 \eta_x^2}{L^2 \eta_y^2} \sum_{t=0}^{T-1} \|\bar{v}_t\|^2 - \frac{12}{5\mu^2} \sum_{t=0}^{T-1} (1 - (1 - \frac{\mu\eta_y}{4})^{T-t}) \|\bar{u}_t\|^2
\end{aligned} \tag{21}$$

*Proof.* Define $z_t = \bar{y}_t + \theta \bar{u}_t$ for some constant $\theta$. As function $f$ is strongly-concave in $y$ we have

$$\begin{aligned}
f(\bar{x}_t, \hat{y}_t) \leq{} & f(\bar{x}_t, \bar{y}_t) + \langle \nabla_y f(\bar{x}_t, \bar{y}_t), \hat{y}_t - \bar{y}_t \rangle - \frac{\mu}{2} \|\hat{y}_t - \bar{y}_t\|^2 \\
={} & f(\bar{x}_t, \bar{y}_t) + \langle \bar{u}_t, \hat{y}_t - z_t \rangle + \langle \nabla_y f(\bar{x}_t, \bar{y}_t) - \bar{u}_t, \hat{y}_t - z_t \rangle \\
& + \theta \langle \nabla_y f(\bar{x}_t, \bar{y}_t), \bar{u}_t \rangle - \frac{\mu}{2} \|\hat{y}_t - \bar{y}_t\|^2
\end{aligned} \tag{22}$$

By Assumption 1, we also have

$$-\frac{L\theta^2}{2} \|\bar{u}_t\|^2 \leq f(\bar{x}_t, z_t) - f(\bar{x}_t, \bar{y}_t) - \theta \langle \nabla_y f(\bar{x}_t, \bar{y}_t), \bar{u}_t \rangle \tag{23}$$

Add Eq. (22) and Eq. (23) together we obtain

$$0 \leq \langle \bar{u}_t, \hat{y}_t - z_t \rangle + \langle \nabla_y f(\bar{x}_t, \bar{y}_t) - \bar{u}_t, \hat{y}_t - z_t \rangle - \frac{\mu}{2} \|\hat{y}_t - \bar{y}_t\|^2 + \frac{L\theta^2}{2} \|\bar{u}_t\|^2 \qquad (24)$$

where we also use the definition of $\hat{y}_t$ so that $f(\bar{x}_t, \hat{y}_t) \geq f(\bar{x}_t, z_t)$.

$$\langle \bar{u}_t, \hat{y}_t - z_t \rangle = -\theta \|\bar{u}_t\|^2 + \langle \bar{u}_t, \hat{y}_t - \bar{y}_t \rangle \qquad (25)$$

Combining Eq. (24) and Eq. (25) we have

$$0 \leq \langle \bar{u}_t, \hat{y}_t - \bar{y}_t \rangle - \frac{\mu}{2} \|\hat{y}_t - \bar{y}_t\|^2 + \langle \nabla_y f(\bar{x}_t, \bar{y}_t) - \bar{u}_t, \hat{y}_t - z_t \rangle - (\theta - \frac{L\theta^2}{2}) \|\bar{u}_t\|^2 \qquad (26)$$

By Cauchy-Schwartz inequality we have

$$\langle \nabla_y f(\bar{x}_t, \bar{y}_t) - \bar{u}_t, \hat{y}_t - z_t \rangle \leq \frac{4}{\mu} \|\nabla_y f(\bar{x}_t, \bar{y}_t) - \bar{u}_t\|^2 + \frac{\mu}{8} \|\hat{y}_t - \bar{y}_t\|^2 + \frac{\mu\theta^2}{8} \|\bar{u}_t\|^2 \qquad (27)$$

Therefore, we obtain

$$0 \leq \langle \bar{u}_t, \hat{y}_t - \bar{y}_t \rangle - \frac{\mu}{4} \|\hat{y}_t - \bar{y}_t\|^2 + \frac{4}{\mu} \|\nabla_y f(\bar{x}_t, \bar{y}_t) - \bar{u}_t\|^2 - (\theta - \frac{L\theta^2}{2} - \frac{\mu\theta^2}{8}) \|\bar{u}_t\|^2$$

$$\leq \langle \bar{u}_t, \hat{y}_t - \bar{y}_t \rangle - \frac{\mu}{4} \|\hat{y}_t - \bar{y}_t\|^2 + \frac{4}{\mu} \|\nabla_y f(\bar{x}_t, \bar{y}_t) - \bar{u}_t\|^2 - \frac{2}{5\mu} \|\bar{u}_t\|^2 \qquad (28)$$

where we let $\theta = \frac{4}{5\mu}$. As we have

$$2\eta_y \langle \bar{u}_t, \hat{y}_t - \bar{y}_t \rangle = \|\bar{y}_t - \hat{y}_t\|^2 + \|\bar{y}_{t+1} - \bar{y}_t\|^2 - \|\bar{y}_{t+1} - \hat{y}_t\|^2 \qquad (29)$$

Eq. (28) is equivalent to

$$\|\bar{y}_{t+1} - \hat{y}_t\|^2 \leq (1 - \frac{\mu\eta_y}{2}) \|\bar{y}_t - \hat{y}_t\|^2 + \|\bar{y}_{t+1} - \bar{y}_t\|^2 + \frac{8\eta_y}{\mu} \|\nabla_y f(\bar{x}_t, \bar{y}_t) - \bar{u}_t\|^2 - \frac{4\eta_y}{5\mu} \|\bar{u}_t\|^2 \qquad (30)$$

When $L\eta_y \leq \frac{1}{5}$, from Eq. (30) we know

$$\|\bar{y}_{t+1} - \hat{y}_t\|^2 \leq (1 - \frac{\mu\eta_y}{2}) \|\bar{y}_t - \hat{y}_t\|^2 + \frac{8\eta_y}{\mu} \|\nabla_y f(\bar{x}_t, \bar{y}_t) - \bar{u}_t\|^2 - \frac{3\eta_y}{5\mu} \|\bar{u}_t\|^2 \qquad (31)$$

According to Young's inequality we have

$$\|\bar{y}_{t+1} - \hat{y}_{t+1}\|^2 \leq (1 + \frac{\mu\eta_y}{4}) \|\bar{y}_{t+1} - \hat{y}_t\|^2 + (1 + \frac{4}{\mu\eta_y}) \|\hat{y}_{t+1} - \hat{y}_t\|^2$$

$$\leq (1 - \frac{\mu\eta_y}{4}) \|\bar{y}_t - \hat{y}_t\|^2 + \frac{9\eta_y}{\mu} \|\nabla_y f(\bar{x}_t, \bar{y}_t) - \bar{u}_t\|^2 + \frac{5\kappa}{L\eta_y} \|\hat{y}_{t+1} - \hat{y}_t\|^2 - \frac{3\eta_y}{5\mu} \|\bar{u}_t\|^2$$

$$\leq (1 - \frac{\mu\eta_y}{4}) \|\bar{y}_t - \hat{y}_t\|^2 + \frac{9\eta_y}{\mu} \|\nabla_y f(\bar{x}_t, \bar{y}_t) - \bar{u}_t\|^2 + \frac{5\kappa^3 \eta_x^2}{L\eta_y} \|\bar{v}_t\|^2 - \frac{3\eta_y}{5\mu} \|\bar{u}_t\|^2 \qquad (32)$$

In the second inequality we use Eq. (31) and $L\eta_y \leq \frac{1}{5}$. The last inequality is because function $y^*(\cdot)$ is $\kappa$-Lipschitz. By Cauchy-Schwartz inequality and Assumption 1 we also have

$$\|\nabla_y f(\bar{x}_t, \bar{y}_t) - \bar{u}_t\|^2 \leq 2\|\bar{u}_t - \frac{1}{n}\sum_{i=1}^n \nabla f_i(x_t^{(i)}, y_t^{(i)})\|^2 + \frac{2L^2}{n}(\|X_t - \bar{X}_t\|_F^2 + \|Y_t - \bar{Y}_t\|_F^2) \qquad (33)$$

Using the definition of $\delta_t$ and the recursion in Eq. (32) we obtain

$$\delta_t \leq (1 - \frac{\mu\eta_y}{4})^t \delta_0 + \frac{9\eta_y}{\mu} \sum_{s=0}^{t-1} (1 - \frac{\mu\eta_y}{4})^{t-s-1} \|\bar{u}_s - \nabla_y f(\bar{x}_t, \bar{y}_t)\|^2 + \frac{5\kappa^3 \eta_x^2}{L\eta_y} \sum_{s=0}^{t-1} (1 - \frac{\mu\eta_y}{4})^{t-s-1} \|\bar{v}_s\|^2$$

$$- \frac{3\eta_y}{5\mu} \sum_{s=0}^{t-1} (1 - \frac{\mu\eta_y}{4})^{t-s-1} \|\bar{u}_s\|^2 \qquad (34)$$

Summing above equation we have

$$\sum_{t=0}^{T-1} \delta_t \leq \frac{4\kappa}{L\eta_y} \delta_0 + \frac{18\eta_y}{\mu} \sum_{t=1}^{T-1} (1 - \frac{\mu\eta_y}{4})^{T-t-1} \sum_{s=0}^{t-1} \|\bar{u}_s - \frac{1}{n}\sum_{i=1}^n \nabla f_i(x_s^{(i)}, y_s^{(i)})\|^2 + \frac{72\kappa^2}{n} \sum_{t=0}^{T-1} (\|X_t - \bar{X}_t\|_F^2$$

$$+ \|Y_t - \bar{Y}_t\|_F^2) + \frac{20\kappa^4 \eta_x^2}{L^2 \eta_y^2} \sum_{t=0}^{T-1} \|\bar{v}_t\|^2 - \frac{12}{5\mu^2} \sum_{t=0}^{T-1} (1 - (1 - \frac{\mu\eta_y}{4})^{T-t}) \|\bar{u}_t\|^2 \qquad (35)$$

where Eq. (33) is used. $\qquad \square$

**Lemma 6.** *For all $t \in \{0, 1, \cdots, T\}$ we have $\bar{v}_t = \bar{g}_t$ and $\bar{u}_t = \bar{h}_t$.*

*Proof.* As matrix $W$ is doubly stochastic, we have

$$\bar{v}_t = \bar{v}_{t-1} + \bar{g}_t - \bar{g}_{t-1} \tag{36}$$

which is equivalent to $\bar{v}_t - \bar{g}_t = \bar{v}_{t-1} - \bar{g}_{t-1}$. Since $\bar{u}_{-1} = \bar{g}_{-1}$, we have $\bar{v}_t = \bar{g}_t$ for all $t \in \{0, 1, \cdots, T\}$. Similarly, we have $\bar{u}_t = \bar{h}_t$. $\qquad\square$

**Lemma 7.** *Let $A_t$, $B_t$ be positive sequences satisfying*

$$A_{t+1} \le (1 - c)A_t + B_t \tag{37}$$

*for some constant $c \in (0, 1)$. Then for any positive integer $T$ we have*

$$\sum_{t=0}^{T} A_t \le \frac{1}{c}A_0 + \frac{1}{c}\sum_{t=0}^{T-1} B_t \tag{38}$$

*Proof.* Using recursion on Eq. (37) we can obtain

$$A_t \le (1 - c)^t A_0 + \sum_{s=0}^{t-1}(1 - c)^{t-s-1}B_s \tag{39}$$

for $\forall t \ge 0$. Sum above inequality and we achieve the desired conclusion Eq. (38), where we use the condtion $A_t$, $B_t$ are positive and the fact that $\sum_{t=0}^{\infty}(1 - c)^t = \frac{1}{c}$. $\qquad\square$

**Lemma 8.** *We can prove the following bound for gradient estimator $\bar{v}_t$ and $\bar{u}_t$.*

$$\sum_{s=0}^{t-1}\mathbb{E}\|\bar{v}_s - \frac{1}{n}\sum_{i=1}^{n}\nabla_x f_i(x_s^{(i)}, y_s^{(i)})\|^2 \le \frac{\sigma^2}{n\beta_x b_0} + \frac{2\beta_x \sigma^2 t}{n} + \frac{12L^2}{n^2 \beta_x}\sum_{s=0}^{t-1}(\mathbb{E}\|X_s - \bar{X}_s\|_F^2 + \mathbb{E}\|Y_s - \bar{Y}_s\|_F^2)$$

$$+ \frac{6L^2}{n\beta_x}\sum_{s=0}^{t-2}(\eta_x^2 \mathbb{E}\|\bar{v}_s\|^2 + \eta_y^2 \mathbb{E}\|\bar{u}_s\|^2) \tag{40}$$

$$\sum_{s=0}^{t-1}\mathbb{E}\|\bar{u}_s - \frac{1}{n}\sum_{i=1}^{n}\nabla_y f_i(x_s^{(i)}, y_s^{(i)})\|^2 \le \frac{\sigma^2}{n\beta_y b_0} + \frac{2\beta_y \sigma^2 t}{n} + \frac{12L^2}{n^2 \beta_y}\sum_{s=0}^{t-1}(\mathbb{E}\|X_s - \bar{X}_s\|_F^2 + \mathbb{E}\|Y_s - \bar{Y}_s\|_F^2)$$

$$+ \frac{6L^2}{n\beta_y}\sum_{s=0}^{t-2}(\eta_x^2 \mathbb{E}\|\bar{v}_s\|^2 + \eta_y^2 \mathbb{E}\|\bar{u}_s\|^2) \tag{41}$$

*for all $t \in \{1, 2, \cdots, T\}$.*

*Proof.* By the definition of $g_t^{(i)}$ and Lemma 6 we have

$$\bar{v}_t - \frac{1}{n}\sum_{i=1}^{n}\nabla_x f_i(x_t^{(i)}, y_t^{(i)})$$

$$= (1 - \beta_x)(\bar{v}_{t-1} - \frac{1}{n}\sum_{i=1}^{n}\nabla_x f_i(x_{t-1}^{(i)}, y_{t-1}^{(i)})) + \frac{\beta_x}{n}\sum_{i=1}^{n}(\nabla_x F_i(x_t^{(i)}, y_t^{(i)}; \xi_t^{(i)}) - \nabla_x f_i(x_t^{(i)}, y_t^{(i)}))$$

$$+ (1 - \beta_x)\frac{1}{n}\sum_{i=1}^{n}\Big(\nabla_x F_i(x_t^{(i)}, y_t^{(i)}; \xi_t^{(i)}) - \nabla_x F_i(x_{t-1}^{(i)}, y_{t-1}^{(i)}; \xi_t^{(i)}) + \nabla_x f_i(x_{t-1}^{(i)}, y_{t-1}^{(i)})$$

$$- \nabla_x f_i(x_t^{(i)}, y_t^{(i)})\Big) \tag{42}$$

Taking expectation on $\xi_t^{(i)}$ the last two terms of Eq. (42) are 0. Therefore,

$$\mathbb{E}\|\bar{v}_t - \frac{1}{n}\sum_{i=1}^{n}\nabla_x f_i(x_t^{(i)}, y_t^{(i)})\|^2$$

$$= (1 - \beta_x)^2 \mathbb{E}\|\bar{v}_{t-1} - \frac{1}{n}\sum_{i=1}^{n}\nabla_x f_i(x_{t-1}^{(i)}, y_{t-1}^{(i)})\|^2 + \mathbb{E}\|\frac{\beta_x}{n}\sum_{i=1}^{n}(\nabla_x F_i(x_t^{(i)}, y_t^{(i)}; \xi_t^{(i)})$$

$$
\begin{aligned}
&- \nabla_x f_i(x_t^{(i)}, y_t^{(i)})) + (1 - \beta_x) \frac{1}{n} \sum_{i=1}^{n} \Big( \nabla_x F_i(x_t^{(i)}, y_t^{(i)}; \xi_t^{(i)}) - \nabla_x F_i(x_{t-1}^{(i)}, y_{t-1}^{(i)}; \xi_t^{(i)}) \\
&\quad + \nabla_x f_i(x_{t-1}^{(i)}, y_{t-1}^{(i)}) - \nabla_x f_i(x_t^{(i)}, y_t^{(i)})) \Big) \|^2 \\
&\leq (1 - \beta_x)^2 \mathbb{E} \|\bar{v}_{t-1} - \frac{1}{n} \sum_{i=1}^{n} \nabla_x f_i(x_{t-1}^{(i)}, y_{t-1}^{(i)}) \|^2 + \frac{2\beta_x^2}{n^2} \sum_{i=1}^{n} \mathbb{E} \| \nabla_x F_i(x_t^{(i)}, y_t^{(i)}; \xi_t^{(i)}) \\
&\quad - \nabla_x f_i(x_t^{(i)}, y_t^{(i)}) \|^2 + \frac{2(1 - \beta_x)^2}{n^2} \sum_{i=1}^{n} \mathbb{E} \| \nabla_x F_i(x_t^{(i)}, y_t^{(i)}; \xi_t^{(i)}) - \nabla_x F_i(x_{t-1}^{(i)}, y_{t-1}^{(i)}; \xi_t^{(i)}) \\
&\quad + \nabla_x f_i(x_{t-1}^{(i)}, y_{t-1}^{(i)}) - \nabla_x f_i(x_t^{(i)}, y_t^{(i)}) \|^2 \\
&\leq (1 - \beta_x)^2 \mathbb{E} \|\bar{v}_{t-1} - \frac{1}{n} \sum_{i=1}^{n} \nabla_x f_i(x_{t-1}^{(i)}, y_{t-1}^{(i)}) \|^2 + \frac{2\beta_x^2 \sigma^2}{n} + \frac{2L^2 (1 - \beta_x)^2}{n^2} (\mathbb{E} \|X_t - X_{t-1}\|_F^2 \\
&\quad + \mathbb{E} \|Y_t - Y_{t-1}\|_F^2)
\end{aligned}
\tag{43}
$$

The first inequality is obtained by Cauchy-Schwartz inequality. In the last inequality we use Lemma 2 on the last two terms and then use Assumption 2, Lemma 1 and Assumption 1. By Cauchy-Schwartz inequality we have estimations

$$
\|X_t - X_{t-1}\|_F^2 \leq 3\|X_t - \bar{X}_t\|_F^2 + 3n\eta_x^2 \|\bar{v}_{t-1}\|^2 + 3\|X_{t-1} - \bar{X}_{t-1}\|_F^2
\tag{44}
$$
$$
\|Y_t - Y_{t-1}\|_F^2 \leq 3\|Y_t - \bar{Y}_t\|_F^2 + 3n\eta_y^2 \|\bar{u}_{t-1}\|^2 + 3\|Y_{t-1} - \bar{Y}_{t-1}\|_F^2
\tag{45}
$$

Combining above two inequalities with Eq. (43) and Lemma 7 we have

$$
\begin{aligned}
&\sum_{s=0}^{t-1} \mathbb{E} \|\bar{v}_s - \frac{1}{n} \sum_{i=1}^{n} \nabla_x f_i(x_s^{(i)}, y_s^{(i)}) \|^2 \\
&\leq \frac{1}{\beta_x} \mathbb{E} \|\bar{v}_0 - \nabla_x f(x_0, y_0)\|^2 + \frac{2\beta_x \sigma^2 t}{n} + \frac{12L^2}{n^2 \beta_x} \sum_{s=0}^{t-1} (\mathbb{E} \|X_s - \bar{X}_s\|_F^2 + \mathbb{E} \|Y_s - \bar{Y}_s\|_F^2) \\
&\quad + \frac{6L^2}{n\beta_x} \sum_{s=0}^{t-2} (\eta_x^2 \mathbb{E} \|\bar{v}_s\|^2 + \eta_y^2 \mathbb{E} \|\bar{u}_s\|^2) \\
&\leq \frac{\sigma^2}{n\beta_x b_0} + \frac{2\beta_x \sigma^2 t}{n} + \frac{12L^2}{n^2 \beta_x} \sum_{s=0}^{t-1} (\mathbb{E} \|X_s - \bar{X}_s\|_F^2 + \mathbb{E} \|Y_s - \bar{Y}_s\|_F^2) \\
&\quad + \frac{6L^2}{n\beta_x} \sum_{s=0}^{t-2} (\eta_x^2 \mathbb{E} \|\bar{v}_s\|^2 + \eta_y^2 \mathbb{E} \|\bar{u}_s\|^2)
\end{aligned}
\tag{46}
$$

for all $t \in \{1, 2, \cdots, T\}$. In the first inequality we use the fact $\frac{1}{1 - (1 - \beta_x)^2} \leq \frac{1}{\beta_x}$ when $\beta_x \leq 1$. The second inequality is because $\mathbb{E} \|\bar{v}_0 - \nabla_x f(x_0, y_0)\|^2 \leq \frac{\sigma^2}{n b_0}$ by Assumption 2 and Lemma 2. Note that if we do not use Lemma 7 on the last term we will get

$$
\begin{aligned}
&\sum_{s=0}^{t-1} \mathbb{E} \|\bar{v}_s - \frac{1}{n} \sum_{i=1}^{n} \nabla_x f_i(x_s^{(i)}, y_s^{(i)}) \|^2 \\
&\leq \frac{\sigma^2}{n\beta_x b_0} + \frac{2\beta_x \sigma^2 t}{n} + \frac{12L^2}{n^2 \beta_x} \sum_{s=0}^{t-1} (\mathbb{E} \|X_s - \bar{X}_s\|_F^2 + \mathbb{E} \|Y_s - \bar{Y}_s\|_F^2) \\
&\quad + \frac{6L^2}{n\beta_x} \sum_{s=0}^{t-2} (1 - (1 - \beta_x)^{t-s-1})(\eta_x^2 \mathbb{E} \|\bar{v}_s\|^2 + \eta_y^2 \mathbb{E} \|\bar{u}_s\|^2)
\end{aligned}
\tag{47}
$$

Mimic above steps we can also prove the second conclusion in Lemma 8. $\qquad \square$

**Lemma 9.** *The consensus error satisfies the following recursive relation*

$$\|X_{t+1} - \bar{X}_{t+1}\|_F^2 \le \frac{1+\lambda^2}{2}\|X_t - \bar{X}_t\|_F^2 + \frac{2\lambda^2\eta_x^2}{1-\lambda^2}\|V_t - \bar{V}_t\|_F^2 \tag{48}$$

$$\|Y_{t+1} - \bar{Y}_{t+1}\|_F^2 \le \frac{1+\lambda^2}{2}\|Y_t - \bar{Y}_t\|_F^2 + \frac{2\lambda^2\eta_y^2}{1-\lambda^2}\|U_t - \bar{U}_t\|_F^2 \tag{49}$$

*Proof.* Let $J = \frac{\mathbf{1}\mathbf{1}^T}{n}$. According to the update rule we have

$$\begin{aligned}
&\|X_{t+1} - \bar{X}_{t+1}\|_F^2 \\
&= \|(X_t - \eta_x V_t)W - (\bar{X}_t - \eta_x \bar{V}_t)\|_F^2 = \|(X_t - \bar{X}_t)(W - J) - \eta_x(V_t - \bar{V}_t)(W - J)\|_F^2 \\
&\le \lambda^2\|X_t - \bar{X}_t\|_F^2 + \lambda^2\eta_x^2\|V_t - \bar{V}_t\|_F^2 - 2\langle(X_t - \bar{X}_t)(W - J), \eta_x(V_t - \bar{V}_t)(W - J)\rangle \\
&\le (\lambda^2 + \theta\lambda^2)\|X_t - \bar{X}_t\|_F^2 + (\frac{\lambda^2\eta_x^2}{\theta} + \lambda^2\eta_x^2)\|V_t - \bar{V}_t\|_F^2 \\
&\le \frac{1+\lambda^2}{2}\|X_t - \bar{X}_t\|_F^2 + \frac{2\lambda^2\eta_x^2}{1-\lambda^2}\|V_t - \bar{V}_t\|_F^2
\end{aligned} \tag{50}$$

In the first inequality we use Assumption 4 and Lemma 3. In the second inequality we use Young's inequality and $\theta$ is an arbitrary positive constant. Let $\theta = \frac{1-\lambda^2}{2\lambda^2}$ and we can get the last inequality. Similar to Eq. (50), we can obtain the following estimation

$$\begin{aligned}
\|Y_{t+1} - \bar{Y}_{t+1}\|_F^2 &= \|(Y_t + \eta_y U_t)W - (\bar{Y}_t + \eta_y \bar{U}_t)\|_F^2 \\
&\le (\lambda^2 + \theta\lambda^2)\|Y_t - \bar{Y}_t\|_F^2 + (\frac{\lambda^2\eta_x^2}{\theta} + \lambda^2\eta_x^2)\|U_t - \bar{U}_t\|_F^2 \\
&\le \frac{1+\lambda^2}{2}\|Y_t - \bar{Y}_t\|_F^2 + \frac{2\lambda^2\eta_y^2}{1-\lambda^2}\|U_t - \bar{U}_t\|_F^2
\end{aligned} \tag{51}$$

$\square$

**Lemma 10.** *For all $t \in \{0, 1, \cdots, T-1\}$ we have*

$$\begin{aligned}
\sum_{s=0}^{t}\mathbb{E}\|V_s - \bar{V}_s\|_F^2 &\le \frac{2}{1-\lambda^2}\mathbb{E}\|V_0 - \bar{V}_0\|_F^2 + \frac{48\lambda^2 L^2}{(1-\lambda^2)^2}\sum_{s=0}^{t}(\mathbb{E}\|X_s - \bar{X}_s\|_F^2 + \mathbb{E}\|Y_s - \bar{Y}_s\|_F^2) \\
&\quad + \frac{24n\lambda^2 L^2}{(1-\lambda^2)^2}\sum_{s=0}^{t-1}\eta_y^2\mathbb{E}\|\bar{u}_s\|^2 + \frac{24n\lambda^2 L^2}{(1-\lambda^2)^2}\sum_{s=0}^{t-1}\eta_x^2\mathbb{E}\|\bar{v}_s\|^2 \\
&\quad + \frac{8\lambda^2\beta_x^2}{(1-\lambda^2)^2}\sum_{s=0}^{t-1}\sum_{i=1}^{n}\mathbb{E}\|g_s^{(i)} - \nabla_x f_i(x_s^{(i)}, y_s^{(i)})\|^2 + \frac{6n\lambda^2\beta_x^2\sigma^2 t}{1-\lambda^2}
\end{aligned} \tag{52}$$

$$\begin{aligned}
\sum_{s=0}^{t}\mathbb{E}\|U_s - \bar{U}_s\|_F^2 &\le \frac{2}{1-\lambda^2}\mathbb{E}\|U_0 - \bar{U}_0\|_F^2 + \frac{48\lambda^2 L^2}{(1-\lambda^2)^2}\sum_{s=0}^{t}(\mathbb{E}\|X_s - \bar{X}_s\|_F^2 + \mathbb{E}\|Y_s - \bar{Y}_s\|_F^2) \\
&\quad + \frac{24n\lambda^2 L^2}{(1-\lambda^2)^2}\sum_{s=0}^{t-1}\eta_y^2\mathbb{E}\|\bar{u}_s\|^2 + \frac{24n\lambda^2 L^2}{(1-\lambda^2)^2}\sum_{s=0}^{t-1}\eta_x^2\mathbb{E}\|\bar{v}_s\|^2 \\
&\quad + \frac{8\lambda^2\beta_y^2}{(1-\lambda^2)^2}\sum_{s=0}^{t-1}\sum_{i=1}^{n}\mathbb{E}\|h_s^{(i)} - \nabla_y f_i(x_s^{(i)}, y_s^{(i)})\|^2 + \frac{6n\lambda^2\beta_y^2\sigma^2 t}{1-\lambda^2}
\end{aligned} \tag{53}$$

*Proof.* By the definition of $V_t$, Assumption 4 and Lemma 3, we have

$$\begin{aligned}
&\|V_{t+1} - \bar{V}_{t+1}\|_F^2 \\
&= \|(V_t + G_{t+1} - G_t)W - (\bar{V}_t + \bar{G}_{t+1} - \bar{G}_t)\|_F^2 \\
&= \|(V_t - \bar{V}_t)(W - J) + (G_{t+1} - G_t)(W - J)\|_F^2 \\
&\le \lambda^2\|V_t - \bar{V}_t\|_F^2 + \lambda^2\|G_{t+1} - G_t\|_F^2 + 2\langle(V_t - \bar{V}_t)(W - J), (G_{t+1} - G_t)(W - J)\rangle
\end{aligned} \tag{54}$$

Review the definition of $g_t^{(i)}$

$$g_{t+1}^{(i)} - g_t^{(i)} = \nabla_x F_i(x_{t+1}^{(i)}, y_{t+1}^{(i)}; \xi_{t+1}^{(i)}) - \nabla_x F_i(x_t^{(i)}, y_t^{(i)}; \xi_{t+1}^{(i)}) - \beta_x(g_t^{(i)} - \nabla_x f_i(x_t^{(i)}, y_t^{(i)}))$$
$$+ \beta_x(\nabla_x F_i(x_t^{(i)}, y_t^{(i)}; \xi_{t+1}^{(i)}) - \nabla_x f_i(x_t^{(i)}, y_t^{(i)})) \tag{55}$$

and take expectation on $\xi_{t+1}^{(i)}$, then we have

$$\mathbb{E}[g_{t+1}^{(i)} - g_t^{(i)}] = \nabla_x f_i(x_{t+1}^{(i)}, y_{t+1}^{(i)}) - \nabla_x f_i(x_t^{(i)}, y_t^{(i)}) - \beta_x(g_t^{(i)} - \nabla_x f_i(x_t^{(i)}, y_t^{(i)})) \tag{56}$$

Taking expectation on $\xi_{t+1}^{(i)}$ the last term of Eq. (54) can be bounded by

$$\mathbb{E}\langle (V_t - \bar{V}_t)(W - J), (G_{t+1} - G_t)(W - J)\rangle$$
$$= \langle (V_t - \bar{V}_t)(W - J), \mathbb{E}[G_{t+1} - G_t](W - J)\rangle \le \lambda\|V_t - \bar{V}_t\|_F \cdot \lambda\|\mathbb{E}[G_{t+1} - G_t]\|_F$$
$$\le \frac{1 - \lambda^2}{4}\|V_t - \bar{V}_t\|_F^2 + \frac{\lambda^4}{1 - \lambda^2}\|\mathbb{E}[G_{t+1} - G_t]\|_F^2$$
$$\le \frac{1 - \lambda^2}{4}\|V_t - \bar{V}_t\|_F^2 + \frac{2\lambda^4}{1 - \lambda^2}\sum_{i=1}^n \|\nabla_x f_i(x_{t+1}^{(i)}, y_{t+1}^{(i)}) - \nabla_x f_i(x_t^{(i)}, y_t^{(i)})\|^2$$
$$+ \frac{2\lambda^4\beta_x^2}{1 - \lambda^2}\sum_{i=1}^n \|g_t^{(i)} - \nabla_x f_i(x_t^{(i)}, y_t^{(i)})\|^2$$
$$\le \frac{1 - \lambda^2}{4}\|V_t - \bar{V}_t\|_F^2 + \frac{2\lambda^4 L^2}{1 - \lambda^2}(\|X_{t+1} - X_t\|_F^2 + \|Y_{t+1} - Y_t\|_F^2)$$
$$+ \frac{2\lambda^4\beta_x^2}{1 - \lambda^2}\sum_{i=1}^n \|g_t^{(i)} - \nabla_x f_i(x_t^{(i)}, y_t^{(i)})\|^2 \tag{57}$$

where the second inequality is resulted from Young's inequality, the third inequality is resulted from Cauchy-Schwartz inequality and the last inequality is resulted from Assumption 1. Besides, applying Cauchy-Schwartz inequality to Eq. (55) we have

$$\mathbb{E}\|g_{t+1}^{(i)} - g_t^{(i)}\|^2$$
$$\le 3\mathbb{E}\|\nabla_x F_i(x_{t+1}^{(i)}, y_{t+1}^{(i)}; \xi_{t+1}^{(i)}) - \nabla_x F_i(x_t^{(i)}, y_t^{(i)}; \xi_{t+1}^{(i)})\|^2 + 3\beta_x^2\mathbb{E}\|g_t^{(i)} - \nabla_x f_i(x_t^{(i)}, y_t^{(i)})\|^2$$
$$+ 3\beta_x^2\mathbb{E}\|\nabla_x F_i(x_t^{(i)}, y_t^{(i)}; \xi_{t+1}^{(i)}) - \nabla_x f_i(x_t^{(i)}, y_t^{(i)})\|^2$$
$$\le 3L^2(\mathbb{E}\|x_{t+1}^{(i)} - x_t^{(i)}\|^2 + \mathbb{E}\|y_{t+1}^{(i)} - y_t^{(i)}\|^2) + 3\beta_x^2\mathbb{E}\|g_t^{(i)} - \nabla_x f_i(x_t^{(i)}, y_t^{(i)})\|^2 + 3\beta_x^2\sigma^2 \tag{58}$$

where in the last inequality we use Assumption 1 and Assumption 2. Combining Eq. (54), (57) and (58) we can obtain

$$\mathbb{E}\|V_{t+1} - \bar{V}_{t+1}\|_F^2 \le \frac{1 + \lambda^2}{2}\mathbb{E}\|V_t - \bar{V}_t\|_F^2 + \frac{4\lambda^2 L^2}{1 - \lambda^2}(\mathbb{E}\|X_{t+1} - X_t\|_F^2 + \mathbb{E}\|Y_{t+1} - Y_t\|_F^2)$$
$$+ \frac{4\lambda^2\beta_x^2}{1 - \lambda^2}\sum_{i=1}^n \mathbb{E}\|g_t^{(i)} - \nabla_x f_i(x_t^{(i)}, y_t^{(i)})\|^2 + 3n\lambda^2\beta_x^2\sigma^2 \tag{59}$$

Then using Eq. (44) and (45) in above inequality we have

$$\mathbb{E}\|V_{t+1} - \bar{V}_{t+1}\|_F^2$$
$$\le \frac{1 + \lambda^2}{2}\mathbb{E}\|V_t - \bar{V}_t\|_F^2 + \frac{12\lambda^2 L^2}{1 - \lambda^2}(\mathbb{E}\|X_{t+1} - \bar{X}_{t+1}\|_F^2 + \mathbb{E}\|Y_{t+1} - \bar{Y}_{t+1}\|_F^2)$$
$$+ \frac{12\lambda^2 L^2}{1 - \lambda^2}(\mathbb{E}\|X_t - \bar{X}_t\|_F^2 + \mathbb{E}\|Y_t - \bar{Y}_t\|_F^2) + \frac{12n\lambda^2 L^2\eta_y^2}{1 - \lambda^2}\mathbb{E}\|\bar{u}_t\|^2$$
$$+ \frac{12n\lambda^2 L^2\eta_x^2}{1 - \lambda^2}\mathbb{E}\|\bar{v}_t\|^2 + \frac{4\lambda^2\beta_x^2}{1 - \lambda^2}\sum_{i=1}^n \mathbb{E}\|g_t^{(i)} - \nabla_x f_i(x_t^{(i)}, y_t^{(i)})\|^2 + 3n\lambda^2\beta_x^2\sigma^2 \tag{60}$$

By Lemma 7, we can further achieve

$$\sum_{s=0}^{t'} \mathbb{E}\|V_s - \bar{V}_s\|_F^2 \le \frac{2}{1 - \lambda^2}\mathbb{E}\|V_0 - \bar{V}_0\|_F^2 + \frac{48\lambda^2 L^2}{(1 - \lambda^2)^2}\sum_{s=0}^{t'}(\mathbb{E}\|X_s - \bar{X}_s\|_F^2 + \mathbb{E}\|Y_s - \bar{Y}_s\|_F^2)$$

$$+ \frac{24n\lambda^2 L^2}{(1-\lambda^2)^2} \sum_{s=0}^{t'-1} \eta_y^2 \mathbb{E}\|\bar{u}_s\|^2 + \frac{24n\lambda^2 L^2}{(1-\lambda^2)^2} \sum_{s=0}^{t'-1} \eta_x^2 \mathbb{E}\|\bar{v}_s\|^2$$

$$+ \frac{8\lambda^2 \beta_x^2}{(1-\lambda^2)^2} \sum_{s=0}^{t'-1} \sum_{i=1}^{n} \mathbb{E}\|g_s^{(i)} - \nabla_x f_i(x_s^{(i)}, y_s^{(i)})\|^2 + \frac{6n\lambda^2 \beta_x^2 \sigma^2 t'}{1-\lambda^2} \tag{61}$$

for all $t' \in \{0, 1, \cdots, T-1\}$. Here we should notice that term $\mathbb{E}\|X_{t+1} - \bar{X}_{t+1}\|_F^2$ in Eq. (60) is summed from $\mathbb{E}\|X_1 - \bar{X}_1\|_F^2$ to $\mathbb{E}\|X_{t'} - \bar{X}_{t'}\|_F^2$, while term $\mathbb{E}\|X_t - \bar{X}_t\|_F^2$ is summed from $\mathbb{E}\|X_0 - \bar{X}_0\|_F^2$ to $\mathbb{E}\|X_{t'-1} - \bar{X}_{t'-1}\|_F^2$. As $X_0 = \bar{X}_0$, these two terms can be merged together. And it is the same with term $\mathbb{E}\|Y_{t+1} - \bar{Y}_{t+1}\|_F^2$. Mimic above steps and we can prove the conclusion for $\sum_{s=0}^{t'} \mathbb{E}\|U_s - \bar{U}_s\|_F^2$ in the similar way. $\qquad \square$

**Lemma 11.** *We can prove the gradient estimators $\bar{g}_t$ and $\bar{h}_t$ satisfy the following conclusion*

$$\sum_{s=0}^{t} \sum_{i=1}^{n} \mathbb{E}\|g_s^{(i)} - \nabla_x f_i(x_s^{(i)}, y_s^{(i)})\|^2 \leq \frac{n\sigma^2}{\beta_x b_0} + 2n\beta_x \sigma^2 t + \frac{12L^2}{\beta_x} \sum_{s=0}^{t} (\mathbb{E}\|X_s - \bar{X}_s\|_F^2 + \mathbb{E}\|Y_s - \bar{Y}_s\|_F^2)$$

$$+ \frac{6nL^2}{\beta_x} \sum_{s=0}^{t-1} (\eta_x^2 \mathbb{E}\|\bar{v}_s\|^2 + \eta_y^2 \mathbb{E}\|\bar{u}_s\|^2) \tag{62}$$

$$\sum_{s=0}^{t} \sum_{i=1}^{n} \mathbb{E}\|h_s^{(i)} - \nabla_y f_i(x_s^{(i)}, y_s^{(i)})\|^2 \leq \frac{n\sigma^2}{\beta_y b_0} + 2n\beta_y \sigma^2 t + \frac{12L^2}{\beta_y} \sum_{s=0}^{t} (\mathbb{E}\|X_s - \bar{X}_s\|_F^2 + \mathbb{E}\|Y_s - \bar{Y}_s\|_F^2)$$

$$+ \frac{6nL^2}{\beta_y} \sum_{s=0}^{t-1} (\eta_x^2 \mathbb{E}\|\bar{v}_s\|^2 + \eta_y^2 \mathbb{E}\|\bar{u}_s\|^2) \tag{63}$$

*for all $t \in \{0, 1, \cdots, T-1\}$.*

*Proof.* According to the definition of $g_t^{(i)}$ we have

$$g_t^{(i)} - \nabla_x f_i(x_t^{(i)}, y_t^{(i)})$$
$$= (1 - \beta_x)(g_{t-1}^{(i)} - \nabla_x f_i(x_{t-1}^{(i)}, y_{t-1}^{(i)})) + \beta_x(\nabla_x F_i(x_t^{(i)}, y_t^{(i)}; \xi_t^{(i)}) - \nabla_x f_i(x_t^{(i)}, y_t^{(i)}))$$
$$+ (1 - \beta_x)\Big(\nabla_x F_i(x_t^{(i)}, y_t^{(i)}; \xi_t^{(i)}) - \nabla_x F_i(x_{t-1}^{(i)}, y_{t-1}^{(i)}; \xi_t^{(i)})$$
$$+ \nabla_x f_i(x_{t-1}^{(i)}, y_{t-1}^{(i)}) - \nabla_x f_i(x_t^{(i)}, y_t^{(i)})\Big) \tag{64}$$

The last two terms of Eq. (64) is 0 after taking expectation of $\xi_t^{(i)}$. Hence we have

$$\mathbb{E}\|g_t^{(i)} - \nabla_x f_i(x_t^{(i)}, y_t^{(i)})\|^2$$
$$= (1 - \beta_x)^2 \mathbb{E}\|g_{t-1}^{(i)} - \nabla_x f_i(x_{t-1}^{(i)}, y_{t-1}^{(i)})\|^2 + \mathbb{E}\|\beta_x(\nabla_x F_i(x_t^{(i)}, y_t^{(i)}; \xi_t^{(i)}) - \nabla_x f_i(x_t^{(i)}, y_t^{(i)}))$$
$$+ (1 - \beta_x)\Big(\nabla_x F_i(x_t^{(i)}, y_t^{(i)}; \xi_t^{(i)}) - \nabla_x F_i(x_{t-1}^{(i)}, y_{t-1}^{(i)}; \xi_t^{(i)})$$
$$+ \nabla_x f_i(x_{t-1}^{(i)}, y_{t-1}^{(i)}) - \nabla_x f_i(x_t^{(i)}, y_t^{(i)})\Big)\|^2$$
$$\leq (1 - \beta_x)^2 \mathbb{E}\|g_{t-1}^{(i)} - \nabla_x f_i(x_{t-1}^{(i)}, y_{t-1}^{(i)})\|^2 + 2\beta_x^2 \mathbb{E}\|\nabla_x F_i(x_t^{(i)}, y_t^{(i)}; \xi_t^{(i)}) - \nabla_x f_i(x_t^{(i)}, y_t^{(i)})\|^2$$
$$+ 2(1 - \beta_x)^2 \mathbb{E}\|\nabla_x F_i(x_t^{(i)}, y_t^{(i)}; \xi_t^{(i)}) - \nabla_x F_i(x_{t-1}^{(i)}, y_{t-1}^{(i)}; \xi_t^{(i)})\|^2$$
$$\leq (1 - \beta_x)^2 \mathbb{E}\|g_{t-1}^{(i)} - \nabla_x f_i(x_{t-1}^{(i)}, y_{t-1}^{(i)})\|^2 + 2\beta_x^2 \sigma^2 + 2(1 - \beta_x)^2 L^2 (\mathbb{E}\|x_t^{(i)} - x_{t-1}^{(i)}\|^2$$
$$+ \mathbb{E}\|y_t^{(i)} - y_{t-1}^{(i)}\|^2) \tag{65}$$

where we use Cauchy-Schwartz inequality and Lemma 1 in the first inequality and use Assumption 1 and Assumption 2 in the last inequality. Sum above inequality from $i = 1$ to $n$ and we have

$$\sum_{i=1}^{n} \mathbb{E}\|g_t^{(i)} - \nabla_x f_i(x_t^{(i)}, y_t^{(i)})\|^2 \leq (1 - \beta_x)^2 \sum_{i=1}^{n} \mathbb{E}\|g_{t-1}^{(i)} - \nabla_x f_i(x_{t-1}^{(i)}, y_{t-1}^{(i)})\|^2 + 2n\beta_x^2 \sigma^2$$

$$+ 2(1 - \beta_x)^2 L^2 (\mathbb{E}\|X_t - X_{t-1}\|^2 + \mathbb{E}\|Y_t - Y_{t-1}\|^2) \tag{66}$$

Then by Eq. (44) and (45) we have

$$\sum_{i=1}^{n} \mathbb{E}\|g_t^{(i)} - \nabla_x f_i(x_t^{(i)}, y_t^{(i)})\|^2$$

$$\leq (1 - \beta_x)^2 \sum_{i=1}^{n} \mathbb{E}\|g_{t-1}^{(i)} - \nabla_x f_i(x_{t-1}^{(i)}, y_{t-1}^{(i)})\|^2 + 2n\beta_x^2 \sigma^2$$

$$+ 6(1 - \beta_x)^2 L^2 (\mathbb{E}\|X_t - \bar{X}_t\|_F^2 + \mathbb{E}\|Y_t - \bar{Y}_t\|_F^2 + \mathbb{E}\|X_{t-1} - \bar{X}_{t-1}\|_F^2$$

$$+ \mathbb{E}\|Y_{t-1} - \bar{Y}_{t-1}\|_F^2) + 6n(1 - \beta_x)^2 L^2 (\eta_x^2 \mathbb{E}\|\bar{v}_{t-1}\|^2 + \eta_y^2 \mathbb{E}\|\bar{u}_{t-1}\|^2) \tag{67}$$

Applying Lemma 7 to Eq. (67), similar to Eq. (61), we can obtain

$$\sum_{s=0}^{t} \sum_{i=1}^{n} \mathbb{E}\|g_s^{(i)} - \nabla_x f_i(x_s^{(i)}, y_s^{(i)})\|^2$$

$$\leq \frac{1}{\beta_x} \sum_{i=1}^{n} \mathbb{E}\|g_0^{(i)} - \nabla_x f_i(x_0^{(i)}, y_0^{(i)})\|^2 + \frac{12L^2}{\beta_x} \sum_{s=0}^{t} (\mathbb{E}\|X_s - \bar{X}_s\|_F^2 + \mathbb{E}\|Y_s - \bar{Y}_s\|_F^2)$$

$$+ \frac{6nL^2}{\beta_x} \sum_{s=0}^{t-1} (\eta_x^2 \mathbb{E}\|\bar{v}_s\|^2 + \eta_y^2 \mathbb{E}\|\bar{u}_s\|^2) + 2n\beta_x \sigma^2 t$$

$$\leq \frac{n\sigma^2}{\beta_x b_0} + 2n\beta_x \sigma^2 t + \frac{12L^2}{\beta_x} \sum_{s=0}^{t} (\mathbb{E}\|X_s - \bar{X}_s\|_F^2 + \mathbb{E}\|Y_s - \bar{Y}_s\|_F^2)$$

$$+ \frac{6nL^2}{\beta_x} \sum_{s=0}^{t-1} (\eta_x^2 \mathbb{E}\|\bar{v}_s\|^2 + \eta_y^2 \mathbb{E}\|\bar{u}_s\|^2) \tag{68}$$

for all $t \in \{0, 1, \cdots, T-1\}$. Here the last inequality is derived by $\mathbb{E}\|g_0^{(i)} - \nabla_x f_i(x_0^{(i)}, y_0^{(i)})\|^2 \leq \frac{\sigma^2}{b_0}$ due to Lemma 2. The estimation of $h_t^{(i)}$ can be achieved in the same way as above. $\qquad \square$

**Lemma 12.** *Let $\eta_x \leq \frac{(1-\lambda)^2}{500L}$ and $\eta_y \leq \frac{(1-\lambda)^2}{500L}$. The consensus error can be bounded by*

$$\sum_{s=0}^{t} (\mathbb{E}\|X_s - \bar{X}_s\|_F^2 + \mathbb{E}\|Y_s - \bar{Y}_s\|_F^2)$$

$$\leq \frac{16\lambda^2 \eta_x^2}{(1-\lambda^2)^3} \mathbb{E}\|V_0 - \bar{V}_0\|_F^2 + \frac{16\lambda^2 \eta_y^2}{(1-\lambda^2)^3} \mathbb{E}\|U_0 - \bar{U}_0\|_F^2 + \frac{576n\lambda^4 L^2 (\eta_x^2 + \eta_y^2)}{(1-\lambda^2)^4} \sum_{s=0}^{t-2} (\eta_x^2 \mathbb{E}\|\bar{v}_s\|^2$$

$$+ \eta_y^2 \mathbb{E}\|\bar{u}_s\|^2) + \frac{64n\lambda^4 (\beta_x \eta_x^2 + \beta_y \eta_y^2)\sigma^2}{(1-\lambda^2)^4 b_0} + \frac{196n\lambda^4 (\beta_x^2 \eta_x^2 + \beta_y^2 \eta_y^2)\sigma^2 t}{(1-\lambda^2)^4} \tag{69}$$

*for all $t \in \{0, 1, \cdots, T\}$.*

*Proof.* Combining Lemma 7 and Lemma 9, for all $t \in \{0, 1, \cdots, T\}$ we have

$$\sum_{s=0}^{t} \|X_s - \bar{X}_s\|_F^2 \leq \frac{4\lambda^2 \eta_x^2}{(1-\lambda^2)^2} \sum_{s=0}^{t-1} \|V_s - \bar{V}_s\|_F^2 \tag{70}$$

Substitute the right side with Lemma 10 we have

$$\sum_{s=0}^{t} \mathbb{E}\|X_s - \bar{X}_s\|_F^2 \leq \frac{8\lambda^2 \eta_x^2}{(1-\lambda^2)^3} \mathbb{E}\|V_0 - \bar{V}_0\|_F^2 + \frac{192\lambda^4 L^2 \eta_x^2}{(1-\lambda^2)^4} \sum_{s=0}^{t-1} (\mathbb{E}\|X_s - \bar{X}_s\|_F^2 + \mathbb{E}\|Y_s - \bar{Y}_s\|_F^2)$$

$$+ \frac{96n\lambda^4 L^2 \eta_x^2}{(1-\lambda^2)^4} \sum_{s=0}^{t-2} \eta_y^2 \mathbb{E}\|\bar{u}_s\|^2 + \frac{96n\lambda^4 L^2 \eta_x^2}{(1-\lambda^2)^4} \sum_{s=0}^{t-2} \eta_x^2 \mathbb{E}\|\bar{v}_s\|^2$$

$$+ \frac{32\lambda^4 \beta_x^2 \eta_x^2}{(1-\lambda^2)^4} \sum_{s=0}^{t-2} \sum_{i=1}^{n} \mathbb{E}\|g_s^{(i)} - \nabla_x f_i(x_s^{(i)}, y_s^{(i)})\|^2 + \frac{24n\lambda^4 \beta_x^2 \eta_x^2 \sigma^2 (t-1)}{(1-\lambda^2)^3} \tag{71}$$

Apply Lemma 11 and we get

$$
\sum_{s=0}^{t} \mathbb{E}\|X_s - \bar{X}_s\|_F^2
$$

$$
\leq \frac{8\lambda^2\eta_x^2}{(1-\lambda^2)^3}\mathbb{E}\|V_0 - \bar{V}_0\|_F^2 + \frac{192\lambda^4 L^2\eta_x^2}{(1-\lambda^2)^4}\sum_{s=0}^{t-1}(\mathbb{E}\|X_s - \bar{X}_s\|_F^2 + \mathbb{E}\|Y_s - \bar{Y}_s\|_F^2)
$$

$$
+ \frac{96n\lambda^4 L^2\eta_x^2}{(1-\lambda^2)^4}\sum_{s=0}^{t-2}(\eta_x^2\mathbb{E}\|\bar{v}_s\|^2 + \eta_y^2\mathbb{E}\|\bar{u}_s\|^2) + \frac{32n\lambda^4\beta_x\eta_x^2\sigma^2}{(1-\lambda^2)^4 b_0} + \frac{64n\lambda^4\beta_x^3\eta_x^2\sigma^2(t-2)}{(1-\lambda^2)^4}
$$

$$
+ \frac{24n\lambda^4\beta_x^2\eta_x^2\sigma^2(t-1)}{(1-\lambda^2)^3} + \frac{384\lambda^4\beta_x L^2\eta_x^2}{(1-\lambda^2)^4}\sum_{s=0}^{t-2}(\mathbb{E}\|X_s - \bar{X}_s\|_F^2 + \mathbb{E}\|Y_s - \bar{Y}_s\|_F^2)
$$

$$
+ \frac{192n\lambda^4\beta_x L^2\eta_x^2}{(1-\lambda^2)^4}\sum_{s=0}^{t-3}(\eta_x^2\mathbb{E}\|\bar{v}_s\|^2 + \eta_y^2\mathbb{E}\|\bar{u}_s\|^2)
$$

$$
\leq \frac{8\lambda^2\eta_x^2}{(1-\lambda^2)^3}\mathbb{E}\|V_0 - \bar{V}_0\|_F^2 + \frac{576\lambda^4 L^2\eta_x^2}{(1-\lambda^2)^4}\sum_{s=0}^{t-1}(\mathbb{E}\|X_s - \bar{X}_s\|_F^2 + \mathbb{E}\|Y_s - \bar{Y}_s\|_F^2)
$$

$$
+ \frac{288n\lambda^4 L^2\eta_x^2}{(1-\lambda^2)^4}\sum_{s=0}^{t-2}(\eta_x^2\mathbb{E}\|\bar{v}_s\|^2 + \eta_y^2\mathbb{E}\|\bar{u}_s\|^2) + \frac{32n\lambda^4\beta_x\eta_x^2\sigma^2}{(1-\lambda^2)^4 b_0} + \frac{98n\lambda^4\beta_x^2\eta_x^2\sigma^2 t}{(1-\lambda^2)^4} \tag{72}
$$

where we use $\beta_x \leq 1$ to simplify the equation. Similarly, we have

$$
\sum_{s=0}^{t} \mathbb{E}\|Y_s - \bar{Y}_s\|_F^2
$$

$$
\leq \frac{8\lambda^2\eta_y^2}{(1-\lambda^2)^3}\mathbb{E}\|U_0 - \bar{U}_0\|_F^2 + \frac{576\lambda^4 L^2\eta_y^2}{(1-\lambda^2)^4}\sum_{s=0}^{t-1}(\mathbb{E}\|X_s - \bar{X}_s\|_F^2 + \mathbb{E}\|Y_s - \bar{Y}_s\|_F^2)
$$

$$
+ \frac{288n\lambda^4 L^2\eta_y^2}{(1-\lambda^2)^4}\sum_{s=0}^{t-2}(\eta_x^2\mathbb{E}\|\bar{v}_s\|^2 + \eta_y^2\mathbb{E}\|\bar{u}_s\|^2) + \frac{32n\lambda^4\beta_y\eta_y^2\sigma^2}{(1-\lambda^2)^4 b_0} + \frac{98n\lambda^4\beta_y^2\eta_y^2\sigma^2 t}{(1-\lambda^2)^4} \tag{73}
$$

Add Eq. (72) and (72) together. Then we have

$$
\sum_{s=0}^{t} (\mathbb{E}\|X_s - \bar{X}_s\|_F^2 + \mathbb{E}\|Y_s - \bar{Y}_s\|_F^2)
$$

$$
\leq \frac{8\lambda^2\eta_x^2}{(1-\lambda^2)^3}\mathbb{E}\|V_0 - \bar{V}_0\|_F^2 + \frac{8\lambda^2\eta_y^2}{(1-\lambda^2)^3}\mathbb{E}\|U_0 - \bar{U}_0\|_F^2 + \frac{576\lambda^4 L^2(\eta_x^2 + \eta_y^2)}{(1-\lambda^2)^4}\sum_{s=0}^{t-1}(\mathbb{E}\|X_s - \bar{X}_s\|_F^2
$$

$$
+ \mathbb{E}\|Y_s - \bar{Y}_s\|_F^2) + \frac{288n\lambda^4 L^2(\eta_x^2 + \eta_y^2)}{(1-\lambda^2)^4}\sum_{s=0}^{t-2}(\eta_x^2\mathbb{E}\|\bar{v}_s\|^2 + \eta_y^2\mathbb{E}\|\bar{u}_s\|^2)
$$

$$
+ \frac{32n\lambda^4(\beta_x\eta_x^2 + \beta_y\eta_y^2)\sigma^2}{(1-\lambda^2)^4 b_0} + \frac{98n\lambda^4(\beta_x^2\eta_x^2 + \beta_y^2\eta_y^2)\sigma^2 t}{(1-\lambda^2)^4} \tag{74}
$$

As $\lambda < 1$, when $\eta_x \leq \frac{(1-\lambda)^2}{500L}$ and $\eta_y \leq \frac{(1-\lambda)^2}{500L}$ it satisfies

$$
\frac{576\lambda^4 L^2(\eta_x^2 + \eta_y^2)}{(1-\lambda^2)^4} \leq \frac{1}{2} \tag{75}
$$

Therefore, Eq. (74) implies

$$\sum_{s=0}^{t}(\mathbb{E}\|X_s - \bar{X}_s\|_F^2 + \mathbb{E}\|Y_s - \bar{Y}_s\|_F^2)$$

$$\leq \frac{16\lambda^2\eta_x^2}{(1-\lambda^2)^3}\mathbb{E}\|V_0 - \bar{V}_0\|_F^2 + \frac{16\lambda^2\eta_y^2}{(1-\lambda^2)^3}\mathbb{E}\|U_0 - \bar{U}_0\|_F^2 + \frac{576n\lambda^4 L^2(\eta_x^2 + \eta_y^2)}{(1-\lambda^2)^4}\sum_{s=0}^{t-2}(\eta_x^2\mathbb{E}\|\bar{v}_s\|^2$$

$$+ \eta_y^2\mathbb{E}\|\bar{u}_s\|^2) + \frac{64n\lambda^4(\beta_x\eta_x^2 + \beta_y\eta_y^2)\sigma^2}{(1-\lambda^2)^4 b_0} + \frac{196n\lambda^4(\beta_x^2\eta_x^2 + \beta_y^2\eta_y^2)\sigma^2 t}{(1-\lambda^2)^4} \tag{76}$$

which reaches the conclusion of Lemma 12. $\qquad\square$

## A.3 Proof for main Theorems

Now we will move forward to the main Theorems in our paper. Here we revise some constant coefficients in the statement, but it does not actually affect the result in our convergence analysis.

**Theorem 3.** *(Restatement of Theorem 1) Let Assumptions 1 to 5 hold. When parameters* $\beta_x = \frac{\epsilon\min\{1,n\epsilon\}}{20}$, $\beta_y = \frac{\epsilon\min\{1,n\epsilon\}}{500\kappa^2}$, $\eta_x = \frac{(1-\lambda)^2\min\{1,n\epsilon\}}{15000\kappa^3 L}$, $\eta_y = \frac{(1-\lambda)^2\min\{1,n\epsilon\}}{1500\kappa L}$, $b_0 = \frac{400\kappa}{\min\{1,n\epsilon\}}$, $T = \frac{30000\kappa^3\epsilon^{-2}}{(1-\lambda)^2\min\{1,n\epsilon\}}$, *our Algorithm 1 satisfies*

$$\frac{1}{T}\sum_{t=0}^{T-1}\mathbb{E}\|\nabla\Phi(\bar{x}_t)\|^2 \leq L(\Phi(x_0) - \Phi^*)\epsilon^2 + \sigma^2\epsilon^2 + L^2\delta_0\epsilon^2 + \frac{\epsilon^2}{n}\sum_{i=1}^{n}\|\nabla_x f_i(x_0, y_0)\|^2$$

$$+ \frac{\epsilon^2}{n}\sum_{i=1}^{n}\|\nabla_y f_i(x_0, y_0)\|^2 \tag{77}$$

*Proof.* Since $\Phi(x)$ is $(\kappa L + L)$-smooth we have

$$\Phi(\bar{x}_t) \leq \Phi(\bar{x}_{t-1}) - \eta_x\langle\bar{v}_{t-1}, \nabla\Phi(\bar{x}_{t-1})\rangle + \eta_x^2\kappa L\|\bar{v}_{t-1}\|^2$$

$$= \Phi(\bar{x}_{t-1}) - \frac{\eta_x}{2}\|\bar{v}_{t-1}\|^2 - \frac{\eta_x}{2}\|\nabla\Phi(\bar{x}_{t-1})\|^2 + \frac{\eta_x}{2}\|\bar{v}_{t-1} - \nabla\Phi(\bar{x}_{t-1})\|^2 + \eta_x^2\kappa L\|\bar{v}_{t-1}\|^2$$

$$\leq \Phi(\bar{x}_{t-1}) - \frac{\eta_x}{2}\|\nabla\Phi(\bar{x}_{t-1})\|^2 - (\frac{\eta_x}{2} - \eta_x^2\kappa L)\|\bar{v}_{t-1}\|^2 + \eta_x\|\bar{v}_{t-1} - \nabla_x f(\bar{x}_{t-1}, \bar{y}_{t-1})\|^2$$

$$+ \eta_x\|\nabla\Phi(\bar{x}_{t-1}) - \nabla_x f(\bar{x}_{t-1}, \bar{y}_{t-1})\|^2 \tag{78}$$

where the last inequality is caused by Cauchy-Schwartz inequality. As we have $\nabla\Phi(\bar{x}_{t-1}) = \nabla_x f(\bar{x}_{t-1}, \hat{y}_{t-1})$, by Assumption 1 the last term satisfies

$$\|\nabla\Phi(\bar{x}_{t-1}) - \nabla_x f(\bar{x}_{t-1}, \bar{y}_{t-1})\|^2 \leq L^2\|\hat{y}_{t-1} - \bar{y}_{t-1}\|^2 = L^2\delta_{t-1} \tag{79}$$

Besides, according to Cauchy-Schwartz inequality we also have

$$\|\bar{v}_{t-1} - \nabla_x f(\bar{x}_{t-1}, \bar{y}_{t-1})\|^2$$

$$\leq 2\|\bar{v}_{t-1} - \frac{1}{n}\sum_{i=1}^{n}\nabla_x f_i(x_{t-1}^{(i)}, y_{t-1}^{(i)})\|^2 + 2\|\frac{1}{n}\sum_{i=1}^{n}\nabla_x f_i(x_{t-1}^{(i)}, y_{t-1}^{(i)}) - \nabla_x f(\bar{x}_{t-1}, \bar{y}_{t-1})\|^2$$

$$\leq 2\|\bar{v}_{t-1} - \frac{1}{n}\sum_{i=1}^{n}\nabla_x f_i(x_{t-1}^{(i)}, y_{t-1}^{(i)})\|^2 + \frac{2L^2}{n}(\|X_{t-1} - \bar{X}_{t-1}\|_F^2 + \|Y_{t-1} - \bar{Y}_{t-1}\|_F^2) \tag{80}$$

Combine Eq. (78), (79), (80) and rearrange the inequality

$$\|\nabla\Phi(\bar{x}_{t-1})\|^2 \leq \frac{2(\Phi(\bar{x}_{t-1}) - \Phi(\bar{x}_t))}{\eta_x} - (1 - 2\kappa L\eta_x)\|\bar{v}_{t-1}\|^2 + 2L^2\delta_{t-1} + \frac{4L^2}{n}(\|X_{t-1} - \bar{X}_{t-1}\|_F^2$$

$$+ \|Y_{t-1} - \bar{Y}_{t-1}\|_F^2) + 4\|\bar{v}_{t-1} - \frac{1}{n}\sum_{i=1}^{n}\nabla_x f_i(x_{t-1}^{(i)}, y_{t-1}^{(i)})\|^2 \tag{81}$$

Telescoping and taking expectation on Eq. (81) we have

$$
\frac{1}{T}\sum_{t=0}^{T-1}\mathbb{E}\|\nabla\Phi(\bar{x}_t)\|^2
$$

$$
\leq \frac{2(\Phi(x_0)-\mathbb{E}\Phi(\bar{x}_T))}{\eta_x T} - \frac{(1-2\kappa L\eta_x)}{T}\sum_{t=0}^{T-1}\mathbb{E}\|\bar{v}_t\|^2 + \frac{2L^2}{T}\sum_{t=0}^{T-1}\mathbb{E}\delta_t + \frac{4L^2}{nT}\sum_{t=0}^{T-1}(\mathbb{E}\|X_t-\bar{X}_t\|_F^2
$$

$$
+ \mathbb{E}\|Y_t-\bar{Y}_t\|_F^2) + \frac{4}{T}\sum_{t=0}^{T-1}\mathbb{E}\|\bar{v}_t - \frac{1}{n}\sum_{i=1}^{n}\nabla_x f_i(x_t^{(i)},y_t^{(i)})\|^2 \tag{82}
$$

Applying Assumption 3, Lemma 5 and Lemma 8 we have

$$
\frac{1}{T}\sum_{t=0}^{T-1}\mathbb{E}\|\nabla\Phi(\bar{x}_t)\|^2
$$

$$
\leq \frac{2(\Phi(x_0)-\Phi^*)}{\eta_x T} - (1-2\kappa L\eta_x - \frac{40\kappa^4\eta_x^2}{\eta_y^2})\frac{1}{T}\sum_{t=0}^{T-1}\mathbb{E}\|\bar{v}_t\|^2 + \frac{8\kappa L^2\delta_0}{TL\eta_y}
$$

$$
+ \frac{148\kappa^2 L^2}{nT}\sum_{t=0}^{T-1}(\mathbb{E}\|X_t-\bar{X}_t\|_F^2 + \mathbb{E}\|Y_t-\bar{Y}_t\|_F^2) + \frac{4}{T}\sum_{t=0}^{T-1}\mathbb{E}\|\bar{v}_t - \frac{1}{n}\sum_{i=1}^{n}\nabla_x f_i(x_t^{(i)},y_t^{(i)})\|^2
$$

$$
+ \frac{36\kappa L\eta_y}{T}\sum_{t=1}^{T-1}(1-\frac{\mu\eta_y}{4})^{T-t-1}\sum_{s=0}^{t-1}\|\bar{u}_s - \frac{1}{n}\sum_{i=1}^{n}\nabla f_i(x_s^{(i)},y_s^{(i)})\|^2
$$

$$
- \frac{24\kappa^2}{5T}\sum_{t=0}^{T-1}(1-(1-\frac{\mu\eta_y}{4})^{T-t})\mathbb{E}\|\bar{u}_t\|^2
$$

$$
\leq \frac{2(\Phi(x_0)-\Phi^*)}{\eta_x T} - (1-2\kappa L\eta_x - \frac{40\kappa^4\eta_x^2}{\eta_y^2})\frac{1}{T}\sum_{t=0}^{T-1}\mathbb{E}\|\bar{v}_t\|^2 + \frac{8\kappa L^2\delta_0}{TL\eta_y} + \frac{4\sigma^2}{nb_0 T}(\frac{1}{\beta_x}+\frac{36\kappa^2}{\beta_y})
$$

$$
+ \frac{8\sigma^2}{n}(\beta_x + 36\kappa^2\beta_y) + \frac{4L^2}{nT}(47\kappa^2 + \frac{12}{n\beta_x} + \frac{432\kappa^2}{n\beta_y})\sum_{t=0}^{T-1}(\mathbb{E}\|X_t-\bar{X}_t\|_F^2 + \mathbb{E}\|Y_t-\bar{Y}_t\|_F^2)
$$

$$
+ \frac{24L^2}{n\beta_x T}\sum_{t=0}^{T-1}(1-(1-\beta_x)^{T-t})(\eta_x^2\mathbb{E}\|\bar{v}_t\|^2 + \eta_y^2\mathbb{E}\|\bar{u}_t\|^2) + \frac{864\kappa^2 L^2}{n\beta_y T}\sum_{t=0}^{T-1}(1-(1-\frac{\mu\eta_y}{4})^{T-t})
$$

$$
\cdot (\eta_x^2\mathbb{E}\|\bar{v}_t\|^2 + \eta_y^2\mathbb{E}\|\bar{u}_t\|^2) - \frac{24\kappa^2}{5T}\sum_{t=0}^{T-1}(1-(1-\frac{\mu\eta_y}{4})^{T-t})\mathbb{E}\|\bar{u}_t\|^2 \tag{83}
$$

where we use Eq. (47) in the last inequality. As

$$
\frac{1}{\beta_x}(1-(1-\beta_x)^{T-t}) = \sum_{s=0}^{T-t-1}(1-\beta_x)^s \tag{84}
$$

we know Eq. (84) is increasing when $\beta_x$ is decreasing. Hence $\frac{1}{\beta_x}(1-(1-\beta_x)^{T-t}) \leq \frac{300\kappa^2}{(1-\lambda)^2\beta_x}(1-(1-\frac{(1-\lambda)^2\beta_x}{300\kappa^2})^{T-t})$. According to the definition of $\beta_x$ and $\eta_y$, we have $\frac{(1-\lambda)^2\beta_x}{300\kappa^2} \leq \frac{\mu\eta_y}{4}$ and

$$
\frac{24L^2}{n\beta_x T}(1-(1-\beta_x)^{T-t}) \leq \frac{7200L^2\kappa^2}{n(1-\lambda)^2\beta_x T}(1-(1-\frac{\mu\eta_y}{4})^{T-t}) \tag{85}
$$

Therefore, using the definition of $\beta_x$, $\beta_y$ and $\eta_y$ we obtain

$$
\frac{1}{T} \sum_{t=0}^{T-1} \mathbb{E}\|\nabla\Phi(\bar{x}_t)\|^2
$$

$$
\leq \frac{2(\Phi(x_0) - \Phi^*)}{\eta_x T} - (1 - 2\kappa L\eta_x - \frac{40\kappa^4\eta_x^2}{\eta_y^2})\frac{1}{T} \sum_{t=0}^{T-1} \mathbb{E}\|\bar{v}_t\|^2 + \frac{8\kappa L^2\delta_0}{TL\eta_y} + \frac{4\sigma^2}{nb_0 T}(\frac{1}{\beta_x} + \frac{36\kappa^2}{\beta_y})
$$

$$
+ \frac{8\sigma^2}{n}(\beta_x + 36\kappa^2\beta_y) + \frac{4L^2}{nT}(47\kappa^2 + \frac{12}{n\beta_x} + \frac{432\kappa^2}{n\beta_y}) \sum_{t=0}^{T-1}(\mathbb{E}\|X_t - \bar{X}_t\|_F^2 + \mathbb{E}\|Y_t - \bar{Y}_t\|_F^2)
$$

$$
+ (\frac{24L^2\eta_x^2}{n\beta_x} + \frac{864\kappa^2 L^2\eta_x^2}{n\beta_y})\frac{1}{T} \sum_{t=0}^{T-1} \mathbb{E}\|\bar{v}_t\|^2 - \frac{\kappa L\eta_y}{T} \sum_{t=0}^{T-1} \mathbb{E}\|\bar{u}_t\|^2 \tag{86}
$$

Besides, according to Lemma 12 we have

$$
\frac{1}{T} \sum_{t=0}^{T-1} \mathbb{E}\|\nabla\Phi(\bar{x}_t)\|^2
$$

$$
\leq \frac{2(\Phi(x_0) - \Phi^*)}{\eta_x T} - (1 - 2\kappa L\eta_x - \frac{40\kappa^4\eta_x^2}{\eta_y^2})\frac{1}{T} \sum_{t=0}^{T-1} \mathbb{E}\|\bar{v}_t\|^2 + \frac{8\kappa L^2\delta_0}{TL\eta_y} + \frac{4\sigma^2}{nb_0 T}(\frac{1}{\beta_x} + \frac{36\kappa^2}{\beta_y})
$$

$$
+ \frac{8\sigma^2}{n}(\beta_x + 36\kappa^2\beta_y) + \frac{4L^2}{nT}(47\kappa^2 + \frac{12}{n\beta_x} + \frac{432\kappa^2}{n\beta_y})\Big(\frac{16\lambda^2\eta_x^2}{(1-\lambda^2)^3}\mathbb{E}\|V_0 - \bar{V}_0\|_F^2
$$

$$
+ \frac{16\lambda^2\eta_y^2}{(1-\lambda^2)^3}\mathbb{E}\|U_0 - \bar{U}_0\|_F^2 + \frac{64n\lambda^4(\beta_x\eta_x^2 + \beta_y\eta_y^2)\sigma^2}{(1-\lambda^2)^4 b_0} + \frac{196n\lambda^4(\beta_x^2\eta_x^2 + \beta_y^2\eta_y^2)\sigma^2 T}{(1-\lambda^2)^4}\Big)
$$

$$
+ \frac{4L^2}{nT}(47\kappa^2 + \frac{12}{n\beta_x} + \frac{432\kappa^2}{n\beta_y})\frac{576n\lambda^4 L^2(\eta_x^2 + \eta_y^2)}{(1-\lambda^2)^4} \sum_{t=0}^{T-1}(\eta_x^2\mathbb{E}\|\bar{v}_t\|^2 + \eta_y^2\mathbb{E}\|\bar{u}_t\|^2)
$$

$$
+ (\frac{24L^2\eta_x^2}{n\beta_x} + \frac{864\kappa^2 L^2\eta_x^2}{n\beta_y})\frac{1}{T} \sum_{t=0}^{T-1} \mathbb{E}\|\bar{v}_t\|^2 - \frac{\kappa L\eta_y}{T} \sum_{t=0}^{T-1} \mathbb{E}\|\bar{u}_t\|^2 \tag{87}
$$

When $\beta_x$, $\beta_y$, $\eta_x$ and $\eta_y$ are defined as Theorem 3, we have

$$
\frac{4L^2}{nT}(47\kappa^2 + \frac{12}{n\beta_x} + \frac{432\kappa^2}{n\beta_y})\frac{576n\lambda^4 L^2(\eta_x^2 + \eta_y^2)}{(1-\lambda^2)^4}\eta_y^2 \leq \frac{\kappa L\eta_y}{2T} \tag{88}
$$

and

$$
1 - 2\kappa L\eta_x - \frac{40\kappa^4\eta_x^2}{\eta_y^2} - \frac{24L^2\eta_x^2}{n\beta_x} - \frac{864\kappa^2 L^2\eta_x^2}{n\beta_y}
$$

$$
- \frac{4L^2}{n}(47\kappa^2 + \frac{12}{n\beta_x} + \frac{432\kappa^2}{n\beta_y})\frac{576n\lambda^4 L^2(\eta_x^2 + \eta_y^2)}{(1-\lambda^2)^4}\eta_x^2 \geq \frac{2}{5} \tag{89}
$$

Thus, we obtain

$$
\frac{1}{T} \sum_{t=0}^{T-1} \mathbb{E}\|\nabla\Phi(\bar{x}_t)\|^2
$$

$$
\leq \frac{2(\Phi(x_0) - \Phi^*)}{\eta_x T} + \frac{8\kappa L^2\delta_0}{TL\eta_y} + \frac{4\sigma^2}{nb_0 T}(\frac{1}{\beta_x} + \frac{36\kappa^2}{\beta_y}) + \frac{8\sigma^2}{n}(\beta_x + 36\kappa^2\beta_y)
$$

$$
+ \frac{4L^2}{nT}(47\kappa^2 + \frac{12}{n\beta_x} + \frac{432\kappa^2}{n\beta_y})\Big(\frac{16\lambda^2\eta_x^2}{(1-\lambda^2)^3}\mathbb{E}\|V_0 - \bar{V}_0\|_F^2 + \frac{16\lambda^2\eta_y^2}{(1-\lambda^2)^3}\mathbb{E}\|U_0 - \bar{U}_0\|_F^2
$$

$$
+ \frac{64n\lambda^4(\beta_x\eta_x^2 + \beta_y\eta_y^2)\sigma^2}{(1-\lambda^2)^4 b_0} + \frac{196n\lambda^4(\beta_x^2\eta_x^2 + \beta_y^2\eta_y^2)\sigma^2 T}{(1-\lambda^2)^4}\Big) \tag{90}
$$

By Assumption 4 and Cauchy-Schwartz inequality we also have

$$\mathbb{E}\|V_0 - \bar{V}_0\|_F^2 = \mathbb{E}\|G_0(W - J)\|_F^2 \leq \lambda^2 \mathbb{E}\|G_0\|_F^2 \leq \frac{2n\lambda^2\sigma^2}{b_0} + 2\lambda^2 \sum_{i=1}^n \|\nabla_x f_i(x_0, y_0)\|^2 \quad (91)$$

Similarly, we have

$$\mathbb{E}\|U_0 - \bar{U}_0\|_F^2 \leq \frac{2n\lambda^2\sigma^2}{b_0} + 2\lambda^2 \sum_{i=1}^n \|\nabla_y f_i(x_0, y_0)\|^2 \quad (92)$$

Combine above three inequalities and substitute the parameters with their definitions. We achieve

$$\frac{1}{T}\sum_{t=0}^{T-1} \mathbb{E}\|\nabla\Phi(\bar{x}_t)\|^2 \leq L(\Phi(x_0) - \Phi^*)\epsilon^2 + L^2\delta_0\epsilon^2 + \sigma^2\epsilon^2 + \frac{\epsilon^2}{n}\sum_{i=1}^n \|\nabla_x f_i(x_0, y_0)\|^2$$

$$+ \frac{\epsilon^2}{n}\sum_{i=1}^n \|\nabla_y f_i(x_0, y_0)\|^2 \quad (93)$$

where we use following inequalities for simplification.

$$\beta_x \geq \beta_y, \frac{144\kappa^2}{n\beta_y b_0 T} \leq \frac{144\kappa^2 \cdot 500\kappa^2(\min\{1, n\epsilon\})^2\epsilon^2}{n\epsilon\min\{1, n\epsilon\}400\kappa \cdot 30000\kappa^3} \leq \frac{3\epsilon^2}{500}$$

$$4L^2(47\kappa^2 + \frac{12}{n\beta_x} + \frac{432\kappa^2}{n\beta_y}) \leq 200L^2\kappa^2 + \frac{1800L^2\kappa^2}{n\beta_y}$$

$$\frac{8\beta_x}{n} \leq \frac{8\epsilon \cdot n\epsilon}{20n} = \frac{2\epsilon^2}{5}, \frac{288\kappa^2\beta_y}{n} \leq \frac{288\kappa^2\epsilon \cdot n\epsilon}{500n\kappa^2} \leq \frac{288\epsilon^2}{500}$$

$$\frac{L^2\beta_x\eta_x^2}{(1-\lambda)^4 b_0 T} \leq \frac{\epsilon(\min\{1, n\epsilon\})^5\epsilon^2}{20 \cdot 400\kappa \cdot 30000\kappa^3(15000\kappa^3)^2}, \frac{L^2\beta_y\eta_y^2}{(1-\lambda)^4 b_0 T} \leq \frac{\epsilon(\min\{1, n\epsilon\})^5\epsilon^2}{500 \cdot 400\kappa \cdot 30000\kappa^3(1500\kappa)^2}$$

$$\frac{L^2\beta_x^2\eta_x^2}{(1-\lambda)^4} \leq \frac{\epsilon^2(\min\{1, n\epsilon\})^4}{400(15000\kappa^3)^2}, \frac{L^2\beta_y^2\eta_y^2}{(1-\lambda)^4} \leq \frac{\epsilon^2(\min\{1, n\epsilon\})^4}{(500\kappa^2)^2(1500\kappa)^2} \quad (94)$$

$\square$

**Theorem 4.** *(Restatement of Theorem 2) Let Assumptions 1 to 5 hold. We set the parameters as* $T = \frac{30000\kappa^3 T_0}{(1-\lambda)^2}$, $\beta_x = \frac{n^{1/3}}{20T_0^{2/3}}$, $\beta_y = \frac{n^{1/3}}{500\kappa^2 T_0^{2/3}}$, $\eta_x = \frac{(1-\lambda)^2 n^{2/3}}{15000\kappa^3 T_0^{1/3} L}$, $\eta_y = \frac{(1-\lambda)^2 n^{2/3}}{1500\kappa T_0^{1/3} L}$, $b_0 = \frac{400\kappa T_0^{1/3}}{n^{2/3}}$, *where we suppose* $T_0 \geq 10n^2$. *Then our algorithm satisfies*

$$\frac{1}{T}\sum_{t=0}^{T-1} \mathbb{E}\|\nabla\Phi(\bar{x}_t)\|^2 \leq \frac{L(\Phi(x_0) - \Phi^*) + \sigma^2 + L^2\delta_0}{(nT_0)^{2/3}} + \frac{\frac{1}{n}\sum_{i=1}^n \mathbb{E}\|\nabla_x f_i(x_0, y_0)\|^2}{T_0}$$

$$+ \frac{\frac{1}{n}\sum_{i=1}^n \mathbb{E}\|\nabla_y f_i(x_0, y_0)\|^2}{T_0} \quad (95)$$

*Proof.* When the parameters are defined as Theorem 4, the conditions in Lemma 5 and Lemma 12 are also satisfied. Hence we can prove Eq. (83) and (87) still hold. When $\beta_x$, $\beta_y$, $\eta_x$ and $\eta_y$ are defined as Theorem 4, we also have

$$\frac{4L^2}{nT}(47\kappa^2 + \frac{12}{n\beta_x} + \frac{432\kappa^2}{n\beta_y})\frac{576n\lambda^4 L^2(\eta_x^2 + \eta_y^2)}{(1-\lambda^2)^4}\eta_y^2 \leq \frac{\kappa L\eta_y}{2T} \quad (96)$$

and

$$1 - 2\kappa L\eta_x - \frac{40\kappa^4\eta_x^2}{\eta_y^2} - \frac{24L^2\eta_x^2}{n\beta_x} - \frac{864\kappa^2 L^2\eta_x^2}{n\beta_y}$$

$$- \frac{4L^2}{n}(47\kappa^2 + \frac{12}{n\beta_x} + \frac{432\kappa^2}{n\beta_y})\frac{576n\lambda^4 L^2(\eta_x^2 + \eta_y^2)}{(1-\lambda^2)^4}\eta_x^2 \geq \frac{2}{5} \quad (97)$$

Similar to Theorem 3, we can also obtain

$$\frac{1}{T}\sum_{t=0}^{T-1}\mathbb{E}\|\nabla\Phi(\bar{x}_t)\|^2$$

$$\leq \frac{2(\Phi(x_0)-\Phi^*)}{\eta_x T} + \frac{8\kappa L^2\delta_0}{TL\eta_y} + \frac{4\sigma^2}{nb_0 T}\Big(\frac{1}{\beta_x} + \frac{36\kappa^2}{\beta_y}\Big) + \frac{8\sigma^2}{n}(\beta_x + 36\kappa^2\beta_y)$$

$$+ \frac{4L^2}{nT}(47\kappa^2 + \frac{12}{n\beta_x} + \frac{432\kappa^2}{n\beta_y})\Big(\frac{16\lambda^2\eta_x^2}{(1-\lambda^2)^3}\mathbb{E}\|V_0 - \bar{V}_0\|_F^2 + \frac{16\lambda^2\eta_y^2}{(1-\lambda^2)^3}\mathbb{E}\|U_0 - \bar{U}_0\|_F^2$$

$$+ \frac{64n\lambda^4(\beta_x\eta_x^2 + \beta_y\eta_y^2)\sigma^2}{(1-\lambda^2)^4 b_0} + \frac{196n\lambda^4(\beta_x^2\eta_x^2 + \beta_y^2\eta_y^2)\sigma^2 T}{(1-\lambda^2)^4}\Big) \tag{98}$$

Substitute the parameters with their definitions and we have

$$\frac{1}{T}\sum_{t=0}^{T-1}\mathbb{E}\|\nabla\Phi(\bar{x}_t)\|^2 \leq \frac{L(\Phi(x_0)-\Phi^*) + \sigma^2 + L^2\delta_0}{(nT_0)^{2/3}} + \frac{\frac{1}{n}\sum_{i=1}^{n}\mathbb{E}\|\nabla_x f_i(x_0, y_0)\|^2}{T_0}$$

$$+ \frac{\frac{1}{n}\sum_{i=1}^{n}\mathbb{E}\|\nabla_y f_i(x_0, y_0)\|^2}{T_0} \tag{99}$$

which achieves the conclusion of Theorem 4. $\qquad\square$