# OpenReview forum: "A Faster Decentralized Algorithm for Nonconvex Minimax Problems"
_NeurIPS.cc/2021/Conference — NeurIPS 2021 Poster_

### Official Review · Reviewer_ptf7 · 2021-07-07

**Rating:** 7
**Confidence:** 5

**Summary:**

This paper proposed the Decentralized Minimax Hybrid Stochastic Gradient Descent (DM-MHGD) algorithm for decentralized stochastic nonconvex-strongly concave minimax optimization with non-iid data. This is the first first-order algorithm with convergence guarantee under this seldom studied setting and it has the optimal SFO complexity $\mathcal{O}(\kappa^3\epsilon^{-3})$ to achieve $\epsilon$-stationary point.

**Ethical Concerns:**

I think this work has no ethical concerns.

**Ethics Review Area:**

["I don’t know"]

**Limitations And Societal Impact:**

Limitations are not mentioned.
Both the authors and I do not think this theoretical work could have any immediate societal impact.

**Main Review:**

Originality: The algorithm and theoretical result are both new for decentralized minimax optimization. The algorithm is a novel combination of techniques including STORM, gradient tracking, and decentralized communication. Related work is adequately cited.

Quality: The submission is technically sound and complete. The claims are well demonstrated by both theoretical analysis and experimental results. The authors did not mention limitations nor future directions.

Clarity: The submission is clear and well organized.

Significance: The results are important as it significantly boosts the convergence decentralized minimax optimization both theoretically and empirically, and thus I think it will likely be cited.

Advice and questions:
(1) In eq. (6), you might change the rightmost “)” into “\big)” or “]” and correspondingly change the left one “(”, which is easier for readers to see the hierarchy of parentheses.

(2) Does $\xi_t^{(i)}$ denote only one sample in Algorithm 1? You could explain that.

(3) About section 3.3:

(3a) Why do the authors pick up SREDA instead of Acc-MDA, etc. to discuss? Both algorithms achieve the optimal SFO sample complexity for centralized stochastic minimax optimization.

(3b) In “SPIDER will probably not converge to a stationary point”, should “SPIDER” be “SREDA”?

(3c) Iterating Eq. (7) yields that $||e_t||^2\le (1-\beta_x)^t ||e_0||^2 + \mathcal{O}(\eta_x^4)/\beta_x$. How can this imply consensus error $||X_t-\overline{X}_t||_F^2=\mathcal{O}(\eta_x^2)$? In addition, both Theorems 1 & 2 use $\eta_x=\mathcal{O}(n\epsilon)$. (Let $\epsilon<<1/n$ and in Theorem 2 let $T_0=\epsilon^{-3}/n$.). In this case, is $\eta_x^2=\mathcal{O}(n^2\epsilon^2)$ not small enough?

(3d) In Example 2, you may change the notations $x_t$ and $y_t$ to the ones that are never used, since I ever confused them with the $x_t$ and $y_t$ in Algorithm 1.

(3e) [2] used diminishing learning rate but you used small constant learning rates in terms of $\epsilon$. Why do you make the change? Is it possible to use larger constant learning rates? Why? You might discuss that.

(4) In Theorem 2, to achieve $\frac{1}{T}\sum_{t=0}^{T-1}\mathbb{E}||\nabla \Phi(\overline{x}_t)||^2\le \epsilon^2$, we need the dominating term $\frac{1}{(nT_0)^{2/3}}=\mathcal{O}(\epsilon^2) \Rightarrow T_0=\epsilon^{-3}/n$, which can be substituted into the hyperparameters and almost recovers Theorem 1. So I think Theorem 2 is enough and Theorem 1 can be a corollary.

(5) In Corollary 2, “linear speedup of our algorithm” is compared with what?

(6) The theorems guarantee the quality of $x_t$. How about that of $y_t$, e. g., $||y_t-y^*(x_t)||$ or $||\nabla_y f(x_t, y_t)||$?

(7) Why are GDmax and Acc-MDA in Table 1 excluded from the experiments? I understand that you have already done many experiments and they look sufficient. Just curious.

(8) In experiment 1, you used over 10000 samples but only 20 devices, how are each $f_j(x, y); j=1, \ldots, 20$ defined?

(9) What are $g_{\theta}$ and $w$ in eq. (14)? At least you could describe their intuitive meaning. Also, is it single-agent RL? If so, usually there is only one reward function and why does each worker node $i$ need to possess a different $R_i$?

(10) Is $\Phi(\cdot)$ in experiment 1 computed from the analytical solution?

(11) You might describe the weights $w_{ij}$ in the ring-based network, such as their values or the distribution that generates them?

(12) You might add the plots of $||\nabla \Phi(x_t)||$ for experiments, possibly in appendix, if convenient.

(13) This following paper analyzes the decentralized variational inequality problem, which generalizes decentralized minimax optimization. Using SGD, it achieves higher complexity than $\mathcal{O}(\epsilon^{-3})$ in non-monotone case (corresponding to non-convex non-concave minimax optimization). You might add it to related work.

Beznosikov, Aleksandr, et al. "Decentralized Local Stochastic Extra-Gradient for Variational Inequalities." arXiv preprint arXiv:2106.08315 (2021).


**Time Spent Reviewing:**

About 4 hours

---

> ### Author Response · Authors · 2021-08-04
> **Response to concerns**
>
> Thank you for your review and acceptance on our paper. We address your concerns as follows.
>
> (1) Thanks for your advice. We will change the parentheses in our final version.
>
> (2) $\xi^{(i)}$ can denote either single sample or a small minibatch of samples.
>
> (3a) Acc-MDA is an extension of STORM so it can be discussed together with STORM. We will point out that it suffers from the same issue when generalizing to decentralized setting in the final version.
>
> (3b) Example 1 is an example of minimization problem which can be regarded as a special minimax problem. Since both SPIDER and SREDA adopt the normalization to update, they encounter the same divergent issue.
>
> (3c) The bound of consensus error $||X_t - \bar{X}_t ||_F^2=O(\eta_x^2)$ is obtain by D-PSGD and it is illustrated by Example 2. In the analysis of STORM, Eq(7) should be satisfied to guarantee the convergence rate. However, according to the definition of $g_t^{(i)}$, we can only obtain
>
> $$|| e_t ||^2 \le (1 - \beta_x) || e_{t-1} ||^2 + O(|| X_t - \bar{X}_t ||_F^2)$$
> Term $|| X_t - \bar{X}_t ||_F^2$ cannot be as small as $O(\eta_x^4)$ in Eq(7).
>
> (3d) Thanks for reminding. We will change the notations.
>
> (3e) Paper [2] proposes two versions of STORM, a diminishing learning rate and a fixed learning rate (in its appendix). But the analysis of these two versions is very similar. For convenience, we adopt the non-adaptive version. The diminishing version is supposed to work in our paper. As we have shown that $|| X_t - \bar{X}_t ||_F^2 = O(\eta_x^2)$, the step should be small because the consensus error should be small when approaching stationary point. Even for flexible learning rate, its final value should be small.
>
> (4) Thanks for the suggestion. We will only keep one main theorem and put another one into the corollary in the final version.
>
> (5) Linear speedup is compared with the single-machine algorithm. We can see how $T_0 = \epsilon^{-3} / n$ is influenced by the number of workers $n$.
>
> (6) On the right side of Eq(81) there is a term of the sum of $\delta_t$. Hence $|| \bar{y}_t – y^*(\bar{x}_t) ||^2$ can also be bounded by the right side of Theorem 1. We will add a remark to show this result.
>
> (7) GDmax is excluded because GT/DA can be treated as its decentralized generalization with gradient tracking acceleration. The reason we do not run Acc-MDA is similar.
>
> (8) The samples are distributed onto the 20 devices. Each local loss function $f_j$ is defined by its own local data with a finite-sum form and the global loss function $f$ is the sum of all local loss function.
>
> (9) Thank you for reminding. The experiment setting follows reference [38] and [39] in our paper. $g_{\theta}$ is the gradient of $V_{\theta}$ and parameter $w$ is yield by Fenchel’s duality. In [39], the paper considers multi-agent RL so the training is decentralized and each worker possesses different $R_i$.
>
> (10) Yes, it has analytical solution since $y^*$ is available.
>
> (11) In the experiment we use uniform weighted ring structure where the weight of each edge is $\frac{1}{3}$ (each diagonal entry of $W$ is also $\frac{1}{3}$).
>
> (12) Thanks for the suggestion. We will add the result in the appendix in our final version.
>
> (13) Thank you for reminding us of the related work.  We will cite and discuss it in our final version.

---

> > ### Comment · Reviewer_ptf7 · 2021-08-21
> > **Reviewer ptf7's 2nd comment**
> >
> > Thank the authors for your reply, which well addresses my concern.
> > You could write down your answer of my question (11) in the experiment section.
> >
> > I also read the other reviewers' comments and the authors' response to them. The authors' response looks good to me. Again, this paper is novel not only in algorithm design, but also proof technique to tackle the challenge caused by the consensus error. **I keep my rating=7.**

---

### Official Review · Reviewer_fAEx · 2021-07-14

**Rating:** 6
**Confidence:** 4

**Summary:**

The paper stuides the nonconvex-strongly-concave min-max optimizaton problem on decentralized setting. In particular, the decentalized min-max problems were seldom studied in the literature and the existing methods suffer from very high gradient complexity. To address this challenge, this paper proposes a new faster decentralized algorithm, namely DM-HSGD, for nonconvex min-max problems by using the variance reduced technique of hybrid stochastic gradient descent. The algorithm achieves SFO complexity of $O(\kappa^3 \epsilon^{-3})$ for decentalized stochastic nonconvex-strongly-concave problems, improving the best existing theoretical results. Numerical results on decentalized settings show the superior performance of DM-HSGD.

**Limitations And Societal Impact:**

See above.

**Main Review:**

The paper provides an interesting combination of techniques for developing arguments for min-max optimization algorithms in decentalized setting and establish the finite convergence guarantee which is new to my knowledge.

Although the paper has a wide range of positive results, it proves these results for a rather restricted class of min-max optimization problems. Specifically, the class in question is nonconvex-strongly-concave min-max problems.

I find this definition to be limited because as far as I understand:
1) This class of min-max problems does not include nonconvex-linear min-max problems which are quite important in practice.
2) The examples from policy evaluation and adversarial training are generally not in this class.

I believe these restrictions, if indeed necessary, should be discussed more clearly in the paper.

Nevertheless, I still believe that this paper makes some interesting contributions to the community. The algorithm is simple and clearly implementable and the convergence analysis seems correct even though I do not check all the details very carefully. One of the technical drawbacks is that the paper leverages several existing techniques from previous papers, e.g., the two-timescale idea from [15] and the STORM framework from [2]. The authors did a great job in comparing STORM and SPIDER (cf. subsection 3.3). However, it is a bit confused why it is nontrivial to combine two-timescale GDA and STORM. Since the results are somehow unsurprising given prior works, a few more sentences on technical contribution seem necessary.

Does this proof carry over to nonconvex-concave min-max problems? A easy approach in my mind is to add a perturbation and reduce nonconvex-concave min-max problems to nonconvex-strongly-concave min-max problems. See the following articles.

T. Lin, C. Jin and M. I. Jordan, "Near-optimal algorithms for minimax optimization", COLT 2020.

Also, the SFO complexity bound of $\tilde{O}(\kappa^{1/2}N\epsilon^{-2})$ can not be achieved by original ProxDIAG and a restarting scheme seems necessary. The above COLT paper and the following article first provide such results in the literature to my knowledge.

D. M. Ostrovskii, A. Lowy, M. Razaviyayn. "Efficient search of first-order Nash equilibria in nonconvex-concave smooth min-max problems". SIOPT 2021.

An additional comment: Assumption 1 imposes Lipschitz gradient condition on stochastic component function and needs to be just justified. Without such condition, it seems difficult to improve $\epsilon^{-4}$ to $\epsilon^{-3}$ and note that the analysis for SGDA requires weaker condition than Assumption 1.

Overall the paper performs a nice mathematical analysis of a new min-max optimization algorithms in decentralized setting by leveraging numerous techniques. Although the setting seems to be a bit restrictive and somewhat similar to previous papers, this paper could be a useful starting point for further follow-up work.

**Time Spent Reviewing:**

3 hours

---

> ### Author Response · Authors · 2021-08-04
> **Response to concerns**
>
> Thank you for your review and acceptance on our paper. We address your concerns as follows.
> 1. Thanks for your suggestion on extending to nonconvex-concave problem. We also think it is promising to explore this case. When we write this paper, we can prove its convergence under nonconvex-concave condition following the outline of nonconvex-concave SGDA. So actually our algorithm can cover the nonconvex-concave and nonconvex-linear problems. However, we notice that the complexity can probably be improved by using accelerated methods like Catalyst and PPA, which are as you mentioned, adding a perturbation to make it strongly concave. Since that will involve much more discussions (if the algorithm is designed differently, the strongly-concave part will also be changed), we decide to only remain the nonconvex-strongly-concave part and plan to study the general concave case as a future work.
> 2. In the policy evaluation experiment in our paper, the $f_i$ is corresponding to the sum of $L_j^{(i)}$ with respect to $j$. It is strongly concave on $w$ if the matrix formed by $g_{\theta}(s_j)$ has full rank. We can assume it is full-rank when the number of samples is much larger than the dimension of $\theta$. The examples of adversarial training and GAN are likely to be nonconvex-concave or even nonconvex-nonconcave, we will study these cases in our future work.
> 3. The Lipschitz gradient assumption for component function is unnecessary in SGDA but is used in the analysis of STORM and SREDA. Besides, it is also used in many decentralized variance reduction algorithms such as D-GET [1] and D-SPIDER-SFO [2].
>
> [1] Improving the Sample and Communication Complexity for Decentralized Non-Convex Optimization: Joint Gradient Estimation and Tracking. (ICML 2020)
>
> [2] D-SPIDER-SFO: A Decentralized Optimization Algorithm with Faster Convergence Rate for Nonconvex Problems. (AAAI 2020)

---

### Official Review · Reviewer_paF7 · 2021-07-15

**Rating:** 6
**Confidence:** 3

**Summary:**

This paper studied the nonconvex-strongly-concave minimax optimization problem on decentralized setting. Existing literature investigateing decentralized minimax problems suffers from high gradient complexity. This paper proposed a faster decentralized algorithm, named as Decentralized Minimax Hybrid Stochastic Gradient Descent (DM-HSGD) and proved that it can achieve SFO complexity of $O( \kappa^3 \epsilon^{-3})$.

**Limitations And Societal Impact:**

There isn't any potential negative societal impact. The limitation/suggestions for improvement have been discussed in the main review above.

**Main Review:**

The main contributions of the paper lies on proposing the DM-HSGD algorithm which is constructed by combining two ingredients: 1. the STORM gradient estimator (from [2]), 2. the gradient tracking that is commonly employed in decentralized optimization (e.g., [4]). The resultant algorithm is a single loop algorithm which achieves a state-of-the-art SFO complexity with a linear speed up property with respect to the number of agents in the network. The study also tested the algorithm numerically on several examples.

The reviewer has the following comments on the current paper in terms of the novelty and the writing quality:

- The algorithm designed merely combines a number of existing techniques to achieve the state-of-the-art complexity result. While the reviewer agrees that this is a new application of the existing techniques to a new optimization problem of interest to the ML community, the paper has failed to illustrate what are the new insights from the application. For example, what are the main challenges in the analysis which are not found in the previous applications? what are the main takeaways from the provided analysis and can the latter be extended to other settings?

- While the main results appear to be accurate (given that they are the applications of existing techniques), there are a number of issues in the writing of the current paper:

-- The reviewer believes that there are missing assumptions regarding the random variable $\xi$ used in Algorithm 1. Particularly, it should consists of unbiased samples drawn such that $E[ \nabla_x F_i( x, y, \xi ) ] = \nabla f_i( x,y, \xi)$. In fact, the distribution of $\xi$ is never defined except for in line 64 where it is mentioned "$\xi^{(i)}$ is an index sampled from the local data", which does not specify the definition of the former.

-- In light of Theorem 1, the statement of Theorem 2 is rather redundant as it is simply a re-cap of the theorem statement of the former. In fact, Corollary 2 suffices to make the point that the proposed algorithm achieves linear speedup.

-- Example 1 and 2 may be presented before the paragraph in line 182 as the two examples are referenced in the main text of Section 3.3.

**----Post-rebuttal----**

Thanks very much for the response. Most of my previous concerns have been addressed and I am happy to adjust my score. However, in the final version, the authors shall (i) elaborate on the challenges of their analysis, (2) fix the issues in presentation as pointed out in the above (e.g., combining Theorem 1 & 2).


**Time Spent Reviewing:**

4

---

> ### Author Response · Authors · 2021-08-04
> **Response to concerns**
>
> Thank you for your review.  We address your concerns as follows.
>
> 1. There are some important challenges that cannot be solved directly using previous works. For example, in decentralized STORM, the consensus error $|| X_t - \bar{X}_t ||_F^2$ is estimated by recursion. However, when it is extended to minimax problem, the term $|| X_t-\bar{X}_t||_F^2$ is affected by $|| Y_t - \bar{Y}_t ||_F^2$, and $|| Y_t - \bar{Y}_t ||_F^2$ also depends on $|| X_t - \bar{X}_t ||_F^2$. Thus, the estimation of the consensus error is different. Besides, the estimation of the variance of gradient is also more challenging. In minimax problem, the last term in Eq.(81) will be influenced by $|| u_t ||$. The bound in HSGD (reference 41 in our paper) is not precise enough in this scenario because the generated $|| u_t ||$ term cannot be compensated by the negative term yield by $\delta_t$ (Lemma 5). In our paper, we have to use a tighter bound of this term that does not treat $1 – (1 - \beta_x)^{T - t}$ as $1$, which is located between line 593 and line 597. This bound can probably be used to other settings where STORM algorithm is applied and there is rigorous requirement on the variance of the gradient. Besides, most of lemmas can be used if the problem is generated to a more general case like nonconvex-concave problem, nonconvex-nonconcave problem or their applications with specific structures.
>
> 2. In stochastic gradient algorithms, we sample a minibatch to calculate the gradient, for example, $v = \frac{1}{b} \sum_{i=1}^b \nabla F(x,y;s_i)$ where $s_i$ are the selected samples. $\xi$ denotes the sample(s) in the minibatch which determines a stochastic gradient estimator. So it is with $\xi^{(i)}$. The specific expression of $D_i$ depends on the specific problem. For example, it can be a uniform distribution with replacement from a certain dataset or data stream. It is true that we assume the stochastic estimator is unbiased, which satisfies $\mathbb{E}_{\xi^{(i)} \sim D_i} \nabla F_i(x,y; \xi^{(i)}) = \nabla f_i(x,y)$. This assumption is widely used and usually used as default. We will point out it clearly in our final version.
>
> 3. Thanks for the suggestion. We will integrate Theorem 1 and Corollary 2 together to make it concise in our final version.
>
> 4. Thank you for reminding. We will change the order of the text and examples in our final version.

---

### Official Review · Reviewer_85Mh · 2021-07-17

**Rating:** 6
**Confidence:** 3

**Summary:**

This paper considers stochastic nonconvex-strongly-concave minimax decentralized optimization problem. The authors propose a new faster decentralized algorithm for solving the considered problem. They showed that the proposed algorithm achieves SFO complexity of $\mathcal{O}(\kappa^3\epsilon^{-3})$, which is the best in the literature for the considered problem. Key to the improved rate is a variance reduction technique and a gradient tracking scheme. Numerical experiments are also presented.

**Limitations And Societal Impact:**

There are no foreseeable negative societal impact of this work.

**Main Review:**

Originality: the authors propose a new faster algorithm for non-convex strongly concave minimax problems. The proposed algorithm is a combination of SPIDER, which aims to accelerate non-convex stochastic optimization, and gradient tracking, which aims to accelerate decentralized optimization.

Quality: the paper is technically sound. All results are supported with solid proof.

Clarification: the presentation of the paper needs to be improved. For example, the definition of SFO complexity should be explicitly introduced. Some insights and impact of the main theorems should be discussed in Section 4.

Significance: the proposed algorithm is a natural combination of some existing algorithms and does not shed much new light on this direction. Overall, the contribution seems rather incremental.

**Time Spent Reviewing:**

4

---

> ### Author Response · Authors · 2021-08-04
> **Response to concerns**
>
> Thank you for your review and acceptance on our paper. We address your concerns as follows.
> 1. Thanks for your advice. We will add the definition of SFO in our final version. We will also add an outline of the proof so that the insight can be easily followed.
>
> 2. Although the techniques exist, applying STORM to decentralized minimax problem is still meaningful because the previous algorithms that achieve the optimal complexity $O(\kappa^3 \epsilon^{-3})$ require large batch size or nested loops, which is not practical in many tasks. Besides, our algorithm is the first stochastic gradient algorithm to solve general decentralized minimax problem on non-identical distributed data with theoretical guarantees.

---

### Decision · Program_Chairs · 2021-09-27

**Decision:**

Accept (Poster)

**Comment:**

All reviewers agree that this paper makes an important contribution by extending recent faster convergence results for nonconvex-strongly concave minimax problems to the decentralized setting and hence I recommend the paper for acceptance. However, I ask the authors to incorporate reviewer suggestions e.g., adding discussion around non-strongly concave problems.